# Temporal evolution of main ambient PM$_{2.5}$ sources in Santiago, Chile, from 1998 to 2012

Francisco Barraza[1,4], Fabrice Lambert[1,4], Héctor Jorquera[2,5], Ana María Villalobos[2], Laura Gallardo[3,4]

[1] Geography Institute, Pontificia Universidad Católica de Chile, Santiago, 7820436, Chile
[2] Department of Chemical Engineering and Bioprocesses, Pontificia Universidad Católica de Chile, Santiago, 7820436, Chile
[3] Department of Geophysics, Universidad de Chile, Santiago, Chile
[4] Center for Climate and Resilience Research, University of Chile, Santiago, Chile
[5] Center for Sustainable Urban Development (CEDEUS), Pontificia Universidad Católica de Chile, Santiago, 7820436, Chile

*Correspondence to*: Francisco Barraza (fjbarraz@uc.cl)

**Abstract.**

The inhabitants of Santiago, Chile have been exposed to harmful levels of air pollutants for decades. The city's poor air quality is a result of steady economic growth, and stable atmospheric conditions adverse to mixing and ventilation that favor the formation of oxidants and secondary aerosols. Identifying and quantifying the sources that contribute to the ambient levels of pollutants is key for designing adequate mitigation measures. Estimating the evolution of source contributions to ambient pollution levels is also paramount to evaluating the effectiveness of pollution reduction measures that have been implemented over the past decades. Here, we quantify the main sources that have contributed to fine particulate matter (PM$_{2.5}$) between April 1998 and August 2012 in downtown Santiago by using two different source-receptor models (PMF 5.0 and UNMIX 6.0), that were applied to elemental measurements of 1243 24-hour filter samples of ambient PM$_{2.5}$. PMF resolved six sources that contributed to ambient PM$_{2.5}$, with UNMIX producing similar results: motor vehicles (37.3±1.1%), industrial sources (18.5±1.3%), copper smelters (14.4±0.8%), wood burning (12.3±1.0%), coastal sources (9.5±0.7%), and urban dust (3.0±1.2%). Our results show that over the 15 years analyzed here, four of the resolved sources significantly decreased [95% Confidence Interval]: motor vehicles 21.3% [2.6, 36.5], industrial sources 39.3% [28.6, 48.4], copper smelters 81.5% [75.5, 85.9], and coastal sources 58.9% [38.5, 72.5], while wood burning didn't significantly change, and urban dust increased by 72% [48.9, 99.9]. These changes are consistent with emission reduction measures, such as improved vehicle emission standards, cleaner smelting technology, introduction of low sulfur diesel for vehicles and natural gas for industrial processes, public transport improvements etc. However, it is also apparent that the mitigation expected from the above regulations has been partially offset by the increasing amount of private vehicle use in the city, with motor vehicles becoming the dominant source of ambient PM$_{2.5}$ in recent years. Consequently, Santiago still experiences ambient PM$_{2.5}$ levels above the annual and 24-hour Chilean and World Health Organization standards, and further regulations are required to reach ambient air quality standards.

## 1 Introduction

Santiago (33.5°S, 70.5°W, 500 m a.s.l.) is the largest metropolitan area in Chile and the 7th in South America, with a population around 7 million. The city is located in a basin confined between a coastal mountain range to the west (height ~ 1000 m a.s.l.) and the Andes range to the east (average height ~ 4000 m a.s.l.) (Figure 1). Moreover, Santiago's climate is controlled by the quasi-permanent influence of the subtropical Pacific high (descending branch of the Hadley's cell), which results in subsidence inversions that inhibit vertical mixing. There is a characteristic radiatively driven circulation that defines up-slope south-westerly winds in the afternoon and down-slope north-easterly winds in the night and morning hours, especially during summer (Muñoz and Undurraga, 2010; Rutllant and Garreaud, 2004; Schmitz, 2005). Sub-synoptic features known as coastal lows recurrently intensify the subsidence conditions (Rutllant and Garreaud, 1995), and their occurrence is linked to acute pollution episodes in winter (Gallardo et al., 2002; Saide et al., 2011). Central Chile is also characterized by significant inter-annual variability connected to El Niño Southern Oscillation (ENSO), and longer-term variability associated with the Pacific Decadal Oscillation (Garreaud et al., 2009). Over the last 6 to 7 years, central and southern Chile has been affected by an extended and persistent drought, partly caused by natural variability and partly linked to a global warming trend (Boisier et al., 2016; CR2, 2015). All these conditions produce favorable conditions for the accumulation of emissions, and the generation of secondary pollutants. The Chilean Meteorological Service does not regularly launch radiosondes in Santiago, so no direct measurements of planetary boundary layer height (PBLH) are available. However, some studies have presented PBLH estimates retrieved from cielometer readings (Muñoz and Undurraga, 2010; Muñoz and Alcafuz, 2012); the data show a distinctive seasonality with lower/higher values for the austral winter/summer seasons, prompted by the synoptic meteorological conditions discussed above, but the PBLH shows no significant trend between 2008 and 2015.

Particulate matter concentrations in Santiago have been recorded according to international standards since the late 1980s (http://sinca.mma.gob.cl/). The evolution of this network in terms of information content has been described elsewhere (Osses et al., 2013; Henriquez et al., 2015;), and several trend analyses have been carried out (Jorquera et al., 2004; Moreno et al., 2010; Mena-Carrasco et al., 2012; Jhun et al., 2013). $PM_{2.5}$ has been monitored in Santiago since 1989, first by the Chilean Ministry of Health, and subsequently by the Chilean Ministry of the Environment, making it one of the longest running $PM_{2.5}$ air quality monitoring networks in the world (Jhun et al., 2013).

Although environmental authorities have gathered a long-term record of ambient $PM_{2.5}$ elemental composition for Santiago, source-apportionment studies are relatively sparse, and they generally include a few months or a single year of data at most (Supplementary table 1). Moreover, they differ methodologically, which makes it hard to infer a trend in source contributions over time. In this study, we provide the first continuous 15-year source-apportionment analysis of ambient $PM_{2.5}$ for Santiago. High concentration levels of $PM_{2.5}$ have been associated with health problems in Santiago (Pino et al., 2004; Cakmak et al., 2007; Valdes et al., 2012; González R. et al., 2013; Leiva G et al., 2013). Since 1990, Chilean authorities have implemented several air pollution abatement polices that have significantly decreased $PM_{2.5}$ (Mena-Carrasco et al., 2014; MMA, 2015). These measures included phasing lead out of gasoline (late nineties), reducing sulfur in diesel fuel for transport and for industry (5000 ppm in 1989 to 15 ppm for vehicular and 50 ppm for industry today), stricter emission standards for mobile sources (from EURO I to EURO III since 2007, EURO IV since 2012), modernization of the public transport fleet, selective bans on private car usage during high pollution days, a mandatory car inspection and maintenance program, street sweeping and cleaning programs, and emissions standards for industrial combustion sources (Sax et al., 2007; Moreno et al., 2010; Jhun et al., 2013;Villalobos et al., 2015). Although these policies have collectively been successful in reducing the occurrence of extreme $PM_{2.5}$ values, annually averaged $PM_{2.5}$ remains well above the World Health Organization (WHO) yearly average guideline of 10 µg/m$^3$ (World Health Organization-WHO, 2005), and above the annual Chilean standard of 20 µg/m$^3$ enacted in 2012 (MMA, 2012). Moreover, Santiago experiences frequent

autumn and winter PM$_{2.5}$ daily episodes with levels exceeding the WHO 24-hour guideline of 25 µg/m$^3$ (WHO, 2005) and the 24-hour Chilean standard of 50 µg/m$^3$. These episodes are recurrent and typically last several days (Saide et al, 2011).

### 1.1 Source apportionment data analyses

Receptor models (see below) are state-of-the-art computational tools that allow researchers to identify and quantify the major sources that contribute to ambient PM$_{2.5}$ concentrations in a given region and over a given period. Within the Latin American region, several source apportionment studies have been carried out in the largest cities such as Mexico City (Mugica et al., 2002), Sao Paulo, Brazil (Andrade et al., 2012), Rio de Janeiro, Brazil (Andrade et al., 2012; Godoy et al., 2009), and Santiago, Chile (Jorquera and Barraza, 2012; Villalobos et al., 2015). However, all these studies spanned only 1 - 2 years, were carried out using different receptor models, and differed in the time period analyzed, so it is difficult to quantitatively compare among them. Nonetheless, traffic and industrial sources are the typical major contributors to ambient PM$_{2.5}$ as shown in Table 1, while biomass burning is relevant only in some cities. The 'other' category source is relevant in most Latin American cities and it may be due to processes leading to organic and inorganic PM$_{2.5}$, plus smaller unresolved sources such as meat cooking, combustion of natural gas, coal, liquefied petroleum gas, etc. (WHO, 2017).

Although these studies provide a quantitative assessment of ambient PM$_{2.5}$ sources, we are aware of no long-term urban source apportionment studies in Latin America. Long-term studies provide a quantitative estimation of the temporal evolution of major contributing sources, so an evaluation of the effectiveness of sector regulations can be performed. This information is critical for policy-makers and stakeholders, to provide feedback and suggest new initiatives to further reduce pollution levels. Table 2 below summarizes several long-term studies carried out in developed and developing countries within a similar period. Motor vehicles and industrial source contributions are clearly higher in developing countries (including most Latin American cities — Table 1), whereas in developed countries those sources have been controlled and their contributions are lower.

### 1.2 Study objectives

Air quality policies and regulations implemented in Santiago were designed using emission inventories that did not include regional sources, such as copper smelters, whose contributions were not explicitly acknowledged in the Air Quality Management plans originally set up in the late 1990s. However, subsequent studies did show the impact of regional sources in Santiago (Gallardo et al., 2002; Olivares et al., 2002), and these industrial sources and the electrical power generation sector have been subject to increasingly stringent emission regulations at the national level in the last two decades. As a result, the relative chemical composition of particles in Santiago has changed with time. However, no study to date has investigated the evolution of ambient particle matter source contributions.

In this study, we seek to identify the major sources in Santiago using elemental characterization for ambient PM$_{2.5}$ filters collected from 1998 to 2012 (1243 samples), analyze how each source varied through time, determine how much each contributed to total ambient PM$_{2.5}$ in Santiago, and how effective the various regulation policies implemented over that period were.

## 2 Methodology

### 2.1 Sampling station

Environmental authorities have collected ambient $PM_{2.5}$ samples in Santiago using 37 mm diameter Teflon filter (Pall Flex) since 1998 using the same sampling and analysis methodology. Filters were collected using a Low-Vol dichotomous samplers (Anderson Instruments, Inc., Smyrna, GA) operating at 15 L/min for 24-hours, with the sampler inlet located 3 meters above ground. The monitoring station is located in Parque O'Higgins, in the interior area of a park in central Santiago, in a relatively flat area of the basin. (Osses et al., 2013) identified it as the most representative site of the Santiago basin. According to other statistical analyses (Gramsch et al., 2006; 2016,), this station can be characterized as an urban background station. Data collected in this station have been used for establishing trends in chemical speciation and source apportionment for particulate matter and epidemiological studies (Koutrakis et al., 2005; Sax et al., 2007; Moreno et al., 2010; Valdes et al., 2012). A total of 1243 daily samples (24-hour filters) were collected about every four days (mean and median) from April 1998 to August 2012

### 2.2 Laboratory and QA/QC analysis

Filters were inspected before being used, and the particles' concentration were determined gravimetrically using a microbalance, with a resolution of 0.01 mg. All filter (blank and filter samples) were stored at constant temperature (22±3°C) and relative humidity (40% HR ±3%) for a least 24-hours before being weighed. Those filters were analyzed using X-ray fluorescence (XRF) at the Desert Research Institute, Reno, NV, USA. The Ministry for the Environment provided the database containing the elemental analyses of those filters. In order to build statistical models based on robust chemical signals, we decided to keep only those elements selected in other studies that used the same data (CMM-MMA, 2011; Koutrakis et al., 2005; Sax et al., 2007; Valdes et al., 2012), for which more than 75% of the samples contained valid measurements above the detection limit. The limit of detection (LOD) was calculated for each element as three times the standard deviation of the blank (blanks represented approximately 10% of the samples). This public database (gravimetry and elemental analysis) has been used in several studies and all of them have already described the laboratory and QA/QC methodology (CMM-MMA, 2011; Jhun et al., 2013; Koutrakis et al., 2005; Sax et al., 2007). Thus, out of the 49 elements reported, 22 agreed with the criteria described above (Na, Mg, Al, Si, S, Cl, K, Ca, Ti, V, Cr, Mn, Fe, Ni, Cu, Zn, As, Se, Br, Sr, Ba and Pb). Out of these, some were discarded because of large data gaps in the time series (Mg, V, Sr, Ba, Se), suspicious sources (Pb and Br, see discussion below), or because the model was not significant (Na and Ca). In the end, 12 elements were used for our analysis: Al, Si, S, Cl, K (as Kns), Ti, Cr, Fe, Ni, Cu, Zn, As. We used two separate methods to address the small missing data gaps in the time series of the selected elements. First, we let the receptor models' (PMF5, UNMIX6) internal algorithms deal with them, which consists of replacing missing values with the median of the complete time-series for each species. Since replacing missing data with the median can lead to severe distortions in the data, we have also used a custom-written algorithm. This method interpolates up to three consecutive missing values using MATLAB's piecewise cubic interpolation algorithm. Sections of four or more consecutive missing values are filled in by summing up a mirrored copy of equal data length on both sides of the missing records, weighted by a $\cos^2$ function, thus ensuring that no artificial frequencies or discontinuities are introduced in the signal — the data filling algorithm is a MATLAB code that is available upon request to F. Lambert. None of the species selected for the receptor model analysis features any data gap larger than 10 data points. Species with larger gaps were ultimately discarded from the analysis. The original and interpolated data are shown in Supplementary figure S1. Since our custom algorithm does not introduce discontinuities in the time series, we used this method for our analysis. In contrast, the receptor model results using the median-based missing data replacement can lose the seasonal signal of some species.

Accordingly, the model results using the median-based filling algorithm yielded more variability in Cl, Ti Cr, Ni and As (species with important number of blanks, see Supplementary figure S1).

### 2.3 Receptor Modeling

Receptor models are mathematical procedures for identifying and quantifying the sources of ambient air pollution and their effects at a receptor site on the basis of concentration measurements, using neither emission inventories, nor meteorological data (Willis, 2000). In mathematical terms, the general receptor modeling problem can be stated in terms of the contributions from $p$ independent sources to $n$ chemical species measured in a set of $m$ samples as follows (Hopke et al., 2006):

$$X_{ij} = \sum_{k=1}^{p} g_{ik} f_{kj} + e_{ij} \tag{1}$$

Where $X_{ij}$ is the j-th species mass measured in the i-th sample, $g_{ik}$ is the PM mass concentration from the k-th source contributing to the i-th sample, $f_{kj}$ is the j-th species mass fraction in the k-th source, $e_{ij}$ is a model residual associated with the j-th species concentration measured in the i-th sample, and $p$ is the total number of independent sources. In this study, we have used two different models from the US Environmental Protection Agency (EPA) to solve the above described equation. The first method is Positive Matrix Factorization (PMF) based on a multivariate factor analysis (Norris et al., 2014). The second method (UNMIX), uses principal component analysis (Norris et al., 2007). The combined use of both methods increases the robustness of our results. The Positive Matrix Factorization (PMF) method, is a multivariate factor analysis tool that decomposes a matrix of speciated sample data into two matrices: factor contributions (G) and factor profiles (F). These factor profiles need to be interpreted by the user to identify the source types that may be contributing to the sample using measured source profile information, emission inventories or key tracer species (Norris et al., 2014). The method is a widely used receptor model for environmental samples (example: indoor and outdoor particulate matter, sediment, wet deposition and surface water) and the theoretical basis and practical implementation issues have been described elsewhere (Reff et al., 2007; Belis et al., 2013) . In this work we have used PMF version 5.0 (Norris et al., 2014) obtained from the EPA website.

The UNMIX method calculates the number of source types, profiles, relative contributions, and a time-series of contributions using sample species concentrations. The species concentrations are apportioned by a principal components analysis using constraints to assure non-negative and realistic source compositions and contributions (Willis, 2000). The theoretical basis and practical implementation issues have been described by Henry (Henry, 2002; 2003). In this work we have used UNMIX version 6.0 (Norris et al., 2007) obtained from the EPA website.

### 2.4 Analysis of source contributions trends

We have used two methods for trend analyses of each source contribution to $PM_{2.5}$. The first is a robust regression to get an evaluation of the long-term change from 1998 to 2012; we used each source contribution in µg/m$^3$ (log transformed to achieve normal distribution) as the dependent variable, and time as the independent variable. The second method detects abrupt transitions in the time series, with the aim to evaluate possible changes due to specific regulations on a particular period. This method uses a Mann-Whitney test with sliding windows of three different lengths (320, 480 and 650 days). We compare the medians of the older and the recent half of the window, and plot the p-value of the hypothesis test result. Low p-values correspond to significant differences between the two halves and therefore a significant change in concentration between those two periods.

## 3 Results and Discussion

### 3.1 Receptor Modeling Results

We have run the two receptor models — PMF 5.0 and UNMIX 6.0 — for different numbers of factors in equation (1) and examined
the resulting source profiles looking for specific tracers and tracer ratios, as well as the seasonality of source contributions to
identify potential sources. We considered 12 species that produced the best model: Al, Si, S, Cl, non-soil K (Kns), Ti, Cr, Fe, Ni,
Cu, Zn and As. This model had the most robust regression parameter and the highest number of statistically significant source
factors. We have discarded Pb and Br in both models, because of the substantial decrease in lead (and bromine) in gasoline and
diesel fuels after 2000, prompted by cleaner fuel policies. Had we included lead in the model, we would have obtained a spurious
source contribution with high values in 1998-2000 and very low values afterwards. This artifact is caused because all receptor
models assume constant chemical composition in the source profiles. Had we kept lead in the model we would have concluded
that the motor vehicle contributions significantly decreased in a span of only one year, which is wrong. Although lead is a classical
tracer of motor vehicle emissions, is still possible to identify and quantify the motor vehicles source using other species or ratios
between species as we did in this work with Cr, Ni, Cu and Zn. To provide a tracer associated with wood burning, we have added
the non-soil potassium parameter Kns calculated as Kns=(K-0.3•Fe) and removed K from the model. The 0.3 coefficient was
obtained from a K-Fe edge plot (supplementary figure 2). This methodology has been used before by Lewis *et. al.* (Lewis et al.,
2003) and Cohen *et. al.* (Cohen et al., 2010) to remove soil contribution from total potassium.

Both models found species regressions with coefficient ($R^2$) greater than 0.7. These species are tracers for the interpretation of
each source profile and of the global model as well (Norris et al., 2007). Then, we applied a multiple linear regression (MLR) to
the daily concentrations of $PM_{2.5}$ using the source contributions {$g_{ik}$} as independent variables, and checked whether the regression
coefficients were positive and statistically significant at the 95% confidence level ($p \leq 0.05$). This approach has been already
described in previous studies (Jorquera and Barraza 2012; Jorquera and Barraza 2013).

Both models produced a similar six-factor solution that explained 74% of the variance in ambient $PM_{2.5}$ (Figure 2). This 74%
figure is a good result if we bear in mind that sources' profiles may not stay constant over the 15-year modeling period, adding
uncertainty to model results. In addition, there is no available data for either organic/elemental carbon or secondary organic aerosols
(SOA). The addition of these two components would have reduced the uncertainty in the resolved source profiles. For instance,
the ratio OC/EC is helpful in discriminating motor vehicles from wood burning. SOA may be relevant in the warm season (October
– April), as shown by Villalobos et al (2015) for Santiago in 2013; these authors estimated that up to 30% of summertime $PM_{2.5}$
may correspond to SOA. Below we discuss each source individually.

The main difference between PMF and UNMIX average results (figure 2) was that UNMIX resolved the oil combustion as a unique
source, apportioning 85% of Ni to that source profile, similar to one found in Santiago by (Artaxo, 1998; Artaxo et al., 1999).
Although vanadium and nickel are good tracers for oil industrial combustion, we had to remove vanadium from the models, because
the high number of missing data in that species precluded a PMF solution. The PMF solution apportions 56.5 and 19.5% of Ni
concentration within the motor vehicles and industrial source profiles, likely because vanadium was excluded from PMF input.
This is a consequence of the different methodologies used by PMF and UNMIX to compute source profiles.

The first source was identified as "motor vehicles", as it contains more than 50% of total Cr, Cu and Zn, which are all tracers of
traffic emissions (Fujiwara et al., 2011). A Zn/Fe ratio of ~ 0.31 can be found in this factor, which is similar to ratios reported in
source apportionment studies in Chilean cities of Temuco, 0.34; Rancagua, 0.31; Iquique, 0.31 (Kavouras et al., 2001) and Las
Condes (0.32) in Santiago (Jorquera and Barraza, 2012) for motor vehicle sources. This source has a characteristic weekly behavior,
with weekend contributions around half the working days' contributions.

The second source was identified as "industrial sources". It is characterized by the high content of sulfur (65.47%) that originates from $SO_2$ emitted by industrial sources. This source also contains other tracer species that originate in industrial processes, such as Ni (19.5%). Since this profile is dominated by sulfur, it has previously also been identified as "industrial sulfates" or "sulfates" (Artaxo et al., 1999; Jorquera and Barraza, 2012; Moreno et al., 2010). This source does not show a weekly change in contribution, as expected from sources that run 24/7.

The third source was identified as "copper smelters". It contains almost all As measured (79%) and its S/As ratio of 23 is close to the values of 17, 15, and 18 obtained in copper smelter profiles resolved by PMF with ambient data from the cities of Rancagua (Kavouras et al., 2001), Quillota (Hedberg et al., 2005), and Las Condes (Santiago) (Jorquera and Barraza, 2012), respectively. Other relevant tracer species found in this source were Cu (21.4 %) and S (18.9%), which were also identified in a previous study on Santiago ( Jorquera and Barraza 2012). There is no significant difference between working days and weekends (p=0.827), which is also consistent with the smelters' continuous operation.

The fourth source was identified as "wood burning". It contains over 70% of non-soil potassium (Kns), suggesting residential wood burning. Also, this source shows an expected seasonal and weekly trend, with winter contributions 5 times higher than during summer, and a working day/weekend ratio of 0.74 (p=$8.28\times10^{-5}$).

The fifth source was identified as "coastal sources". These are coastal aerosols that reach Santiago's basin. This source contains 90% of the Cl, suggesting a strong sea salt component, present all year long (Jorquera and Barraza, 2012). It also contains minor contributions of industrial sources, such as Ni (8.6%), Zn (9.9%) and As (4.7%), which suggest a contribution from anthropogenic coastal emissions as well. This source shows no weekly cycle (p=0.251), as expected for natural or continuous anthropogenic sources.

Finally, the sixth source was identified as "urban dust". It contains most of Al, Si, and Ti and features elemental ratios that indicate soil dust contribution (Malm et al., 1994). For example, its Si/Al ratio of 2.26 is close to source apportionment results from other Chilean cities (Temuco, 2.17; Valparaíso, 2.58) (Kavouras et al., 2001). Also, this source did not present any contribution of the pseudo species Kns as expected; this validates that using Kns was a proper choice to discriminate soil from wood burning. This source shows a significant higher contribution during working days, which can be explained by the higher number of vehicles on the street during workdays that resuspend street dust (ratio working day/weekend = 1.18; p=$3.53\times10^{-4}$).

We ran UNMIX 6.0 using the same data selected for the PMF 5.0 calculations and obtained similar results (Figure 2). The main difference is that we could not identify the source "industrial sources" using UNMIX 6.0, because the PMF solution apportions sulfur among the sources "urban dust", "coastal sources", "wood burning", and "copper smelters". Instead, UNMIX resolves a source that we identified as "oil combustion", with a high contribution of Ni and Cr and low values of Cu, Zn and As. This source has been identified in previous studies as a contributor to Santiago's ambient $PM_{2.5}$ ( Rojas et al., 1990; Artaxo, 1996; 1998; Jhun et al., 2013). This 'oil combustion' source was not resolved by PMF, as discussed above.

The unexplained source concentration can be calculated by the intercept value in both models. For the PMF 5.0 model the unexplained fraction represents 5% of mass, but it is not statistically significant (intercept estimate has a p value of 0.052) and could therefore be a statistical artifact. For the UNMIX 6.0 model the unexplained fraction was statistically significant at 7% of $PM_{2.5}$ mass (intercept estimate has a p value of 0.0046). This unexplained fraction could be due to local sporadic or secondary sources such as secondary organic aerosols (Villalobos et al, 2015). The average contributions of both models are shown in figure 2.

### 3.2 Mass concentration and seasonal behavior

Over the whole study period, the daily mean (24-hour) concentration of $PM_{2.5}$ was 35.60 (standard deviation 27.89) $\mu g/m^3$ and the
median 24.19 $\mu g/m^3$ (median absolute deviation of 11.81) —figure 3. Over the 15 years of the analyzed time period, there were 599
and 293 days when daily $PM_{2.5}$ concentrations were above WHO and Chilean standards, respectively. The highest daily levels are
found during the cold seasons (autumn and winter) with a ratio close to 3 between cold seasons and warm season concentrations
(Table 3). During the spring and summer seasons, boundary layer height increases (Muñoz and Undurraga, 2010; Muñoz and
Alcafuz, 2012) along with wind speeds. The seasonal signal in most sources is strongly linked with the reduction of air volume
below the boundary layer during the cold season, and does not necessarily imply seasonal variability in the emissions. The
contribution of all sources to $PM_{2.5}$ air concentrations increases during the cold season, with almost all episodes with $PM_{2.5}$ levels
over the Chilean and WHO standards occurring during autumn and winter.

The six identified sources have distinct seasonal contributions to $PM_{2.5}$. During winter, when $PM_{2.5}$ reaches its largest
concentrations and ambient temperatures are lowest, we find a maximum contribution from residential wood smoke to total $PM_{2.5}$
because of the widespread use of fire wood for heating purposes. On the other hand, the other five sources have their highest
contributions during autumn. Unlike wood burning, the emissions from these sources are more constant through the seasons and
their contribution to $PM_{2.5}$ concentrations more strongly modulated by meteorological conditions. Their peak during the latter half
of the cold season is explained by the rainy season that takes place during winter and washed out a large fraction of the contaminants
from the air. The lack of rainfall and low boundary layer height during autumn produces the maximum contributions in these
sources.

### 3.3 Time series of each source contribution

#### 3.3.1 Motor Vehicles

In Figure 4 we show the temporal evolution of the source identified by PMF as motor vehicles. Over the 15 years analyzed in this
study, the motor vehicles contribution to $PM_{2.5}$ decreased significantly by 2.17 ($\pm$1.91) $\mu g/m^3$, p=0.0250 (21.3%, 95% CI [2.6%,
36.5%]) This is explained by several policy measures: restrictions to vehicle traffic since late 1980s (Moreno et al., 2010),
mandatory catalytic converters for gasoline powered cars since 1991 (Koutrakis et al., 2005), reduction of sulfur in gasoline and
diesel, operation of new urban highways and the implementation of a new public transportation system between 2007 and 2010
called "Transantiago" (Muñoz et al., 2014).

The reduction in motor vehicles' contributions has not been constant. In 2002 there was a reduction of 1.53 $\mu g/m^3$ (15.6%), which
is due to the improvement of gasoline and diesel quality, highlighted by the fact that lead was entirely removed from gasoline on
April 2001. (Moreno et al., 2010). Between 2005 and 2006, four new urban highways opened, and one of these is only 2.4 km
from the sampling site. During this period, the vehicles contribution increased in 1.6 $\mu g/m^3$ (18.9%). We ascribe this rise to the
proximity of the new highways to the sampling site as well as the increase in the traffic there. Brudgge et.. al., concluded over
several studies in several cities that proximity to highways increase the exposition to air particles (Brudgge et al., 2007).

In February 2007, a new fully integrated public transport system for Santiago ("Transantiago") was implemented. One of its goals
was to reduce atmospheric emissions, thus improving air quality in the city. However, the motor vehicles contribution increased
by 1.34 $\mu g/m^3$ (13.4%) during 2007-2008. Unfortunately, the beginning of Transantiago included design flaws, bad operation, and
chaotic implementation (Muñoz et al., 2014). In addition, the bus fleet was drastically reduced from ~ 8000 to ~4500 buses in early
2007. This reduced fleet was insufficient to cope with demand and – compounded with the above problems with Transantiago –
induced people to buy and use private cars, which led to an 11% increase in the motorized vehicle fleet in 2007 (S3 and S4).

Subsequent improvements in the Transantiago public transport system after 2007 led to a reduced contribution of motor vehicles. The measures included i) a renewed increase of the bus fleet to 6000 vehicles to satisfy passenger demand, ii) an extension of the subway network, and iii) the gradual implementation of EURO III emission standards for buses (from 53% of the fleet in 2007 to 92% in May 2012 (Muñoz et al., 2014)). Comparing the period 2010-2011 with 2005-2006 we found a long-term decrease of motor vehicles contribution of 2.78 µg/m$^3$ (27.7%) that can be ascribed to Transantiago's full implementation.

### 3.3.2 Industrial sources

In Figure 5 we show the temporal evolution of the source identified as industrial sources. This source reduced its contributions from 1998 to 2012 by 2.63 (±0.71) µg/m$^3$, p=1.1×10$^{-9}$ (39.3%, 95% CI [28.6, 48.4%]). This improvement can be explained by abatement policies for sulfur in industrial diesel fuel (Jhun et al., 2013), mandatory reductions in industrial emissions (Mena-Carrasco et al., 2014), and a change from diesel to natural gas as industrial fuel (MMA 2015). We found a significant reduction of 2.52 µg/m$^3$ (34.3%) in 2002 compared with 2001, that can be explained by a reduction of sulfur in industrial diesel, which was reduced from 1000 to 300 ppm in 2001 (CMM, 2014; MMA, 2015).

Between 2005 and 2007 we find a significant increase of industrial sources contributions, which was triggered by the phasing out of natural gas imports from Argentina. During those years, a large number of industries were forced to switch back to diesel fuel or fuel oil. Since 2008, the Chilean government imports liquefied natural gas from other countries, which explains the reduction between 2009 and 2010 caused by industries switching back to natural gas (Figueroa et al., 2013; GNL-Quitero, 2016). The period 2010-2012 shows a reduction of 1.76 µg/m (31.2%) compared with the period without natural gas imports (2004-2008).

### 3.3.3 Copper Smelters

In Figure 6 we show the temporal evolution of the source identified as copper smelters. This contribution features the largest reduction of 5.24 (±1.38), µg/m$^3$, p=0.82×10$^{-33}$ (81.5%, 95% CI [75.8%, 85.9%]) between 1998 and 2012. This decrease can be attributed to technological improvements at the Caletones and Ventanas smelters near Santiago (see figure 1). In 1998, new regulations forced Caletones to install an acid plant for $SO_2$ abatement, then a second one in 2002 (Minsegpres, 1998). This emission abatement technology decreased $SO_2$ emissions from 700,000 tons in 1999 to 100,000 tons in 2003 (CODELCO, 2015; Montezuma, 2016). The period between 2002 and 2010 shows a reduction of 4.13 µg/m$^3$ (69.0%) compared to the period 1998-2001. We find another significant reduction of 1.41 µg/m$^3$ (64.7 %) between 2009 and 2012 explained by further reductions in $SO_2$ emissions at both smelters. The Ventanas smelter reduced its $SO_2$ emissions from 20.3 kton/year in 2009 to 4.7 kton/year in 2012, while the Caletones smelter's $SO_2$ emissions were reduced from 141 kton/year in 2009 to 50 kton/year in 2012 (Montezuma, 2016).

### 3.3.4 Wood Burning

In Figure 7 we show the temporal evolution of the source identified as wood burning. It is the only resolved source with no net significant change in the period 1998-2012: 0.43 (±0.60) µg/m$^3$, p=0.139 (12.8%, 95% CI [-4.9%, 27.6%]). Nevertheless, we find two significant changes during this period that canceled each other out: i) an increase in 2007-2009 of 1.17 µg/m$^3$ (43.4%), compared with 2004-2006, and ii) a reduction in 2010-2012 of 1.16 µg/m$^3$ (30.1%) compared with 2007-2009.

To curb down wood burning emissions, Chilean authorities have forbidden open chimneys since 1997. Only allow the use of certified woodstoves. In addition, residential wood burning is completely banned during bad air quality episodes (Mena-Carrasco et al., 2012). Our results show that these measures have not been effective (at least during the studied period) to reduce wood burning contribution to PM$_{2.5}$. Mena-Carrasco at al., 2012 suggested the replacement of current wood stoves in Santiago with stoves using cleaner fuels as a cost-effective way of reducing air pollution. They estimated a reduction of 2.07 µg/m$^3$ in PM$_{2.5}$

concentrations if all wood stoves were changed to natural gas stoves. This would represent about 50% of our estimate of current wood burning contributions to Santiago's ambient $PM_{2.5}$.

**3.3.5 Coastal sources**

In Figure 8 we show the temporal evolution of the source identified by PMF as coastal sources. This source is a mixture of marine aerosols and coastal industry emission. Its contribution shows a significant reduction of 1.48 (±0.51) µg/m$^3$, p=0.88×10$^{-5}$ (58.9%, 95% CI [38.5%, 72.5%]), which we attribute to changes in the coastal industry, while we assume that marine aerosols remained constant. We find a significant reduction of 1.62 µg/m$^3$ (77.5%) from 2000 to 2002 compared with the period 1998-1999 that can

be explained by cleaner industrial fuel, as explained for the industrial source contribution case. Likewise, those coastal industries were also affected by the shortage of natural gas imports from Argentina, increasing their contributions from 2004 – 2008 due to a temporary switch to diesel, and then reducing contributions after 2009 following LNG imports. Since 2010, coastal sources have reduced their contribution by 1.05 µg/m$^3$ (76.2%) compared with the period 2007-2008.

**3.3.6 Urban Dust**

In Figure 9 we show the temporal evolution of the source identified as urban dust. It is the only resolved source that has increased its contributions significantly by 0.49 (±0.18) µg/m$^3$, p = 0.26×10$^{-12}$ (72.6%, 95% CI [48.9%, 99.9%]) from 1998 to 2012. Three significant changes are apparent. The first is a reduction of 0.42 µg/m$^3$ (48.8%) between 2001 and 2002, which can be explained by the improvement of the fuel quality in 2001, when lead was removed from gasoline (Jhun et al., 2013). Ayrault et al., 2013 showed that lead particles emitted by gasoline can be deposited on surface soil and remain there for a long time.

A second change was an increase of 0.67 µg/m$^3$ (171.8%) from 2004 to 2010, which can be explained by the significant increase of the number of cars in the city (Instituto Nacional de Estadísticas, 2016). The third significant change was in 2011, with an increase of 0.48 µg/m$^3$ (51.61%). Two factors may explain this rise: i) an annual increase of 7% in the number of cars (Instituto Nacional de Estadísticas, 2016), and ii) since 2010 central Chile has experienced an extended drought (Boisier et al., 2016; CR2, 2015), which leads to drier conditions and promotes aeolian aerosol resuspension.

In Table 4 we summarize the main changes in each source contribution and the corresponding air quality regulation or other events that caused these changes.

**3.3.7 Relative changes in source contributions**

From 1998 to 2012 total $PM_{2.5}$ concentrations have been reduced as a consequence of the air quality regulations described above.

However, individual sources did not vary in the same proportion, and their relative contributions changed over the 15 years (Figure 10). The main reduction was effected for copper smelter contributions that reduced their relative contribution to total $PM_{2.5}$ from 33% in 1998-1999 to 5% in 2011-2012. On the other hand, the impact of motor vehicles increased significantly in relative terms, to the point that this source has become the largest $PM_{2.5}$ contributor since 2003-2005. In connection with the rise in motor vehicle numbers after 2005, the 73% increase in urban dust raised its contribution to $PM_{2.5}$ from 3% to 7%.

One should note that during the time period discussed here, precipitation decreased in central Chile (Supplementary figure 5), leading to a worsening of dispersion conditions in Santiago's basin during autumn and winter. Therefore, the estimated changes for the four sources that decreased their contributions are a lower bound estimate of the reductions in the respective source emissions. Likewise, the relative increase in urban dust estimated is an upper bound of actual dust emission changes.

## 4. Conclusions

We have applied two different receptor models (PMF 5.0 and UNMIX 6.0) to a multiyear database of ambient $PM_{2.5}$ concentrations measured in air filters (1243 samples) collected in a central site in Santiago, Chile. Both models resolve six major sources of ambient $PM_{2.5}$ (motor vehicles, industrial sources, copper smelters, wood burning, coastal sources, and urban dust) and show the temporal evolution of each source from 1998 to 2012. Five of the six identified sources feature significant seasonal variability, increasing their contributions during autumn and winter, and triggering a high number of episodes with harmful concentrations of

$PM_{2.5}$.

During the 15 years analyzed in this study (1998-2012) several air quality regulations were implemented by regional authorities, with the aim of reducing ambient particle levels in Santiago. The most successful measures were those that targeted industrial sources, particularly the regulation of copper smelter emissions and the introduction of cleaner fuels. Copper smelters, coastal sources and industrial sources reduced their contribution by 5.24 (±1.38), 1.48 (±0.51), and 2.63 (±0.71) µg/m$^3$, respectively, or

380 81.5%, 39.3% and 58.9%, respectively, from 1998 to 2012. These estimates are lower bounds of the respective changes in emissions sources, because of a steady decrease in precipitation during this time period (Supplementary figure 5).

Motor vehicles contribution was reduced by 2.17 (±1.91) µg/m$^3$ (21.3%) over the whole period, again a lower bound estimate in traffic emissions changes. Although vehicle fleets have moved to cleaner technologies, the fast growth in the number of private cars has offset part of the gains achieved from tighter vehicle emission standards. Thus, a big challenge for the future is the

385 promotion of behavioral changes in commuters to choose public transportation or non-motorized travel over private cars.

Urban dust (a mixture of crustal and road dust) is the only identified source that has significantly increased its contribution to total $PM_{2.5}$. Our estimated 0.49 (±0.18), µg/m$^3$ (72.6%) increase since 1998 is likely an upper bound in dust emissions changes. It might be due to the rise in private vehicle trips over the years, leading to road dust suspension, combined with drier conditions in central Chile as experienced since 2010 (Supplementary figure 5). Its overall contribution to $PM_{2.5}$ was nevertheless minor (< 1.41 µg/m$^3$

or < 10 % of total $PM_{2.5}$) in 2012, in agreement with long-term source apportionment studies elsewhere (Table 2).

We did not find any significant long-term change in residential wood burning contributions. This source is particularly important in the cold season when it account for roughly 30.6 % of $PM_{2.5}$. Measures to reduce this source's contribution are urgently needed to improve winter air quality in Santiago. However, the road to achieve such reduction is not an easy one: cultural tradition and risk misperception are barriers for change in household practices (Hine et al. 2007; Reeve et al. 2013).

Although government measures have been successful at improving air quality over the past decades, the inhabitants of Santiago are still exposed to harmful $PM_{2.5}$ concentrations that stay above Chilean ambient standards and WHO guidelines for a significant amount of time. It is apparent that industrial sources have been capped significantly. Besides further industrial emission reductions, our study suggests that policies aimed at reducing traffic and residential emissions should be emphasized, as there is still a large reduction potential for these two sources. Table 2 shows that in developed countries with similar climate it is feasible to achieve

source contributions that are substantially lower than the current estimates for Santiago.

## 5 Acknowledgments

Financial support for this work was provided by grants FONDECYT 3160639 and 1151427, FONDAP 15110009 and 15110020, and Anillo ACT1410. We thank the Chilean Ministry of Environment for providing $PM_{2.5}$ filter analysis data. The data filling algorithm is a MATLAB code that is available upon request to F. Lambert.

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

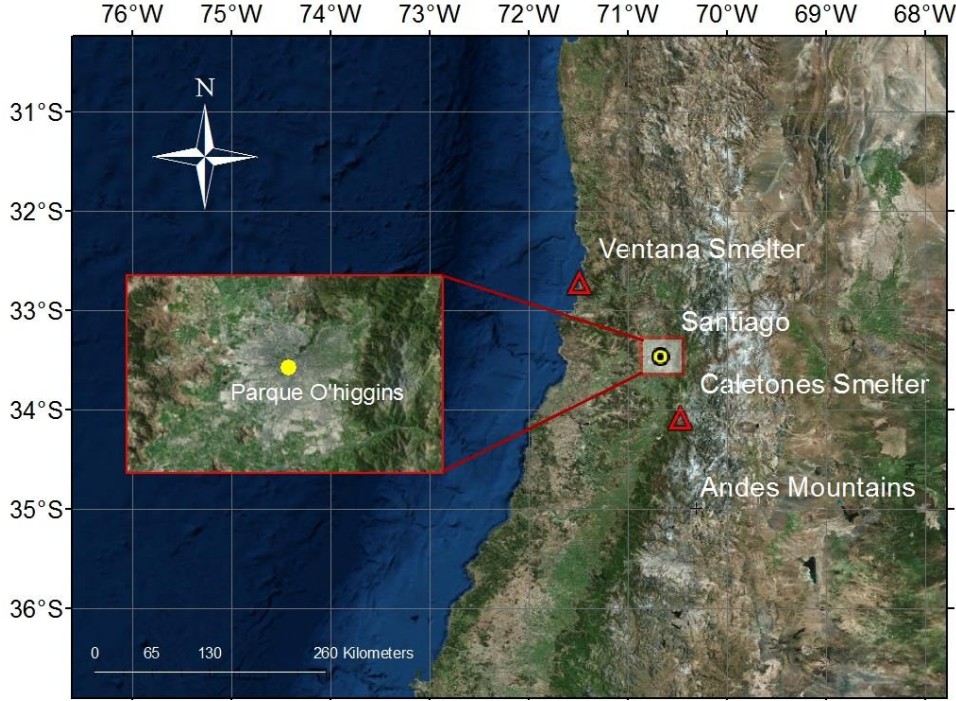

**Figure 1 Map of Santiago region, Chile, with the metropolitan area indicated by the red rectangle, and the yellow circle showing the location of the monitoring site in Parque O'Higgins. The red triangles show the location of the major copper smelters close to Santiago.**

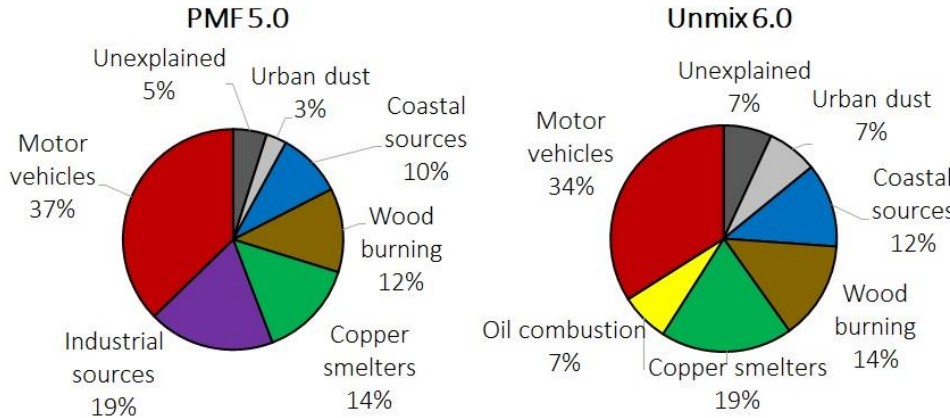

**Figure 2 Source apportionment of fine particulate matter in Santiago, Chile, over the whole period 1998-2012 using two different models. The PM2.5 median over 15 years was 24.19 µg/m3.**

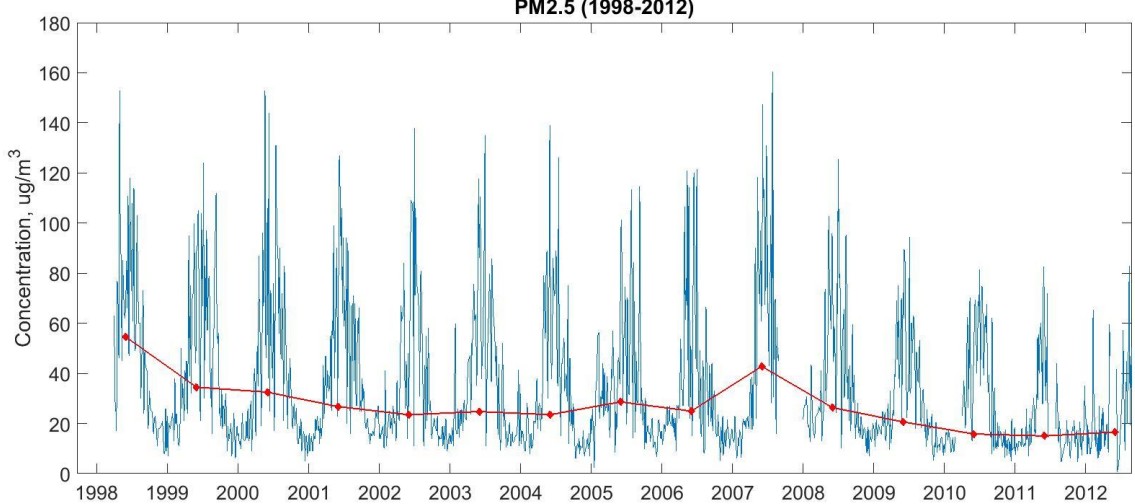

**Figure 3 Temporal evolution of PM2.5 concentrations in Parque O'Higgins monitoring station in central Santiago. The red line shows the annual median.**

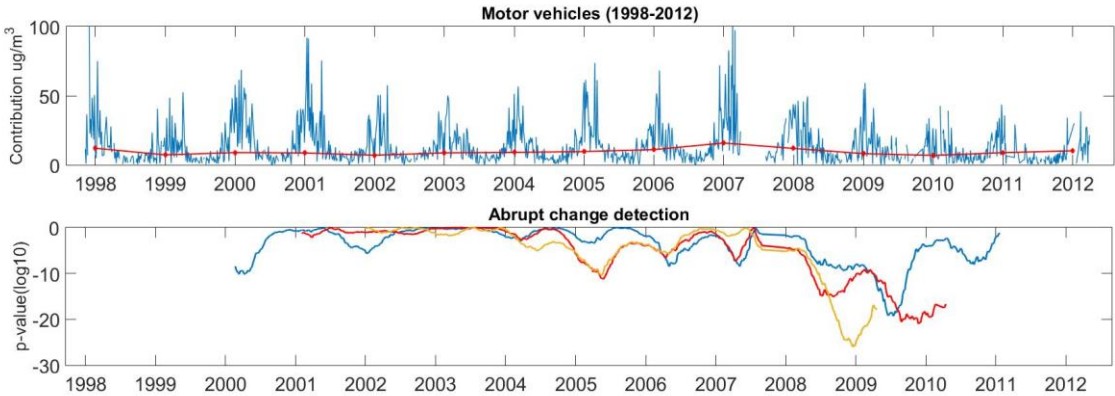

**Figure 4 Top panel: Time series of motor vehicles contribution to PM2.5 and the annual median in red. Bottom panel: p-value from a Mann-Whitney hypothesis test comparing the medians of both halves of a sliding window, repeated for 3 different windows lengths (320, 480 and 640 days for blue, red and yellow, respectively).**

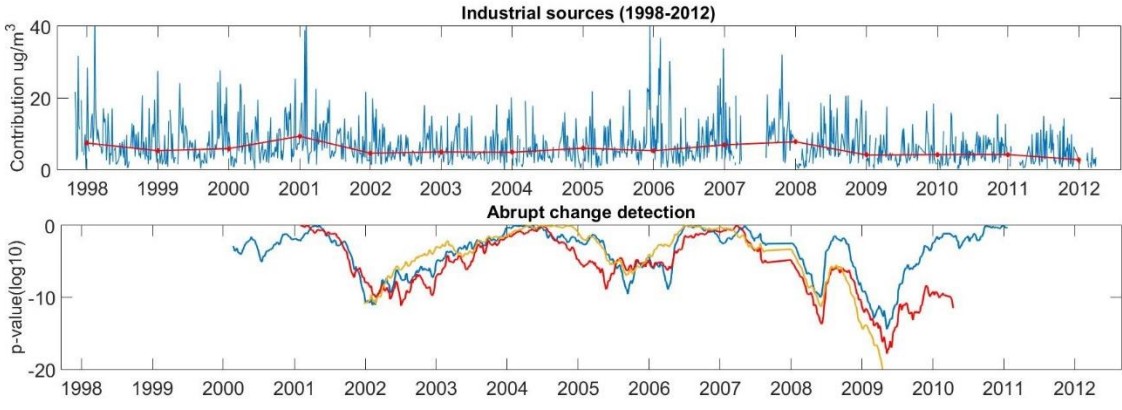

**Figure 5 Top panel: Time series of industrial sources contribution to PM2.5 and the annual median in red. Bottom panel: p-value from a Mann-Whitney hypothesis test comparing the median of both halves of a sliding window, repeated for 3 different windows lengths (320, 480 and 640 days for blue, red and yellow, respectively).**

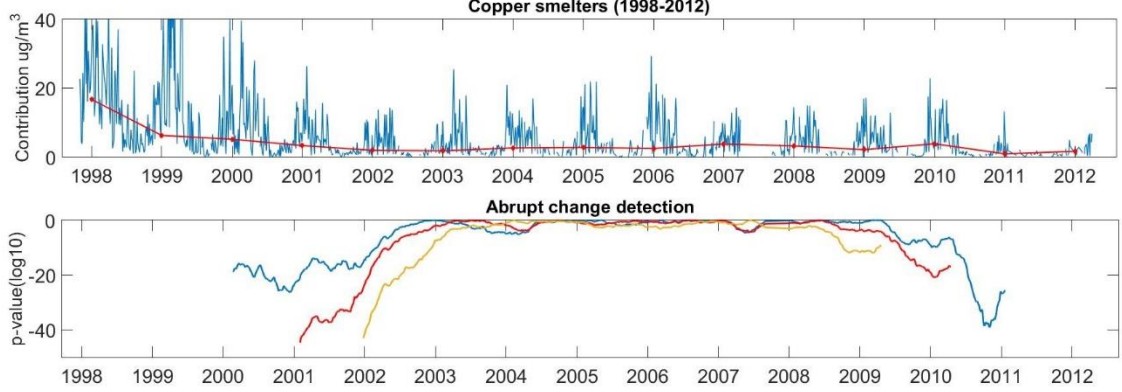

**Figure 6 Top panel: Time series of copper smelters contribution to PM2.5 and the annual median in red. Bottom panel: p-value from a Mann-Whitney hypothesis test comparing the median of both halves of a sliding window, repeated for 3 different windows lengths (320, 480 and 640 days for blue, red and yellow, respectively).**

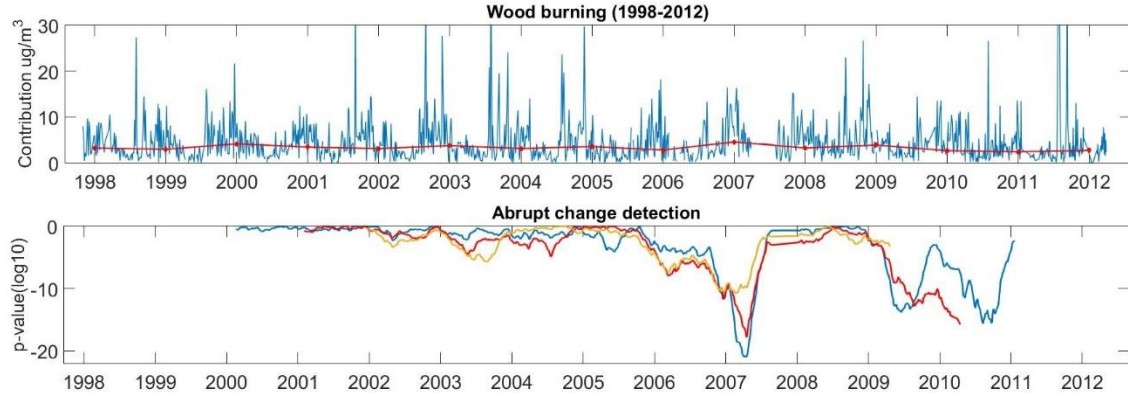

**Figure 7 Top panel: Time series of wood burning contribution to PM2.5 and the annual median in red. Bottom panel: p-value from a Mann-Whitney hypothesis test comparing the median of both halves of a sliding window, repeated for 3 different windows lengths (320, 480 and 640 days for blue, red and yellow, respectively).**

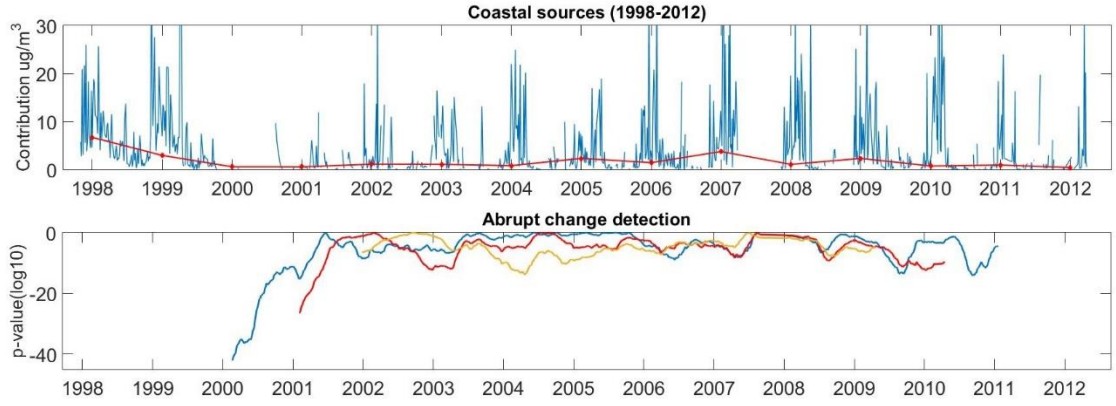

**Figure 8 Top panel: Time series of coastal sources contributions to PM2.5 and the annual median in red. Bottom panel: p-value from a Mann-Whitney hypothesis test comparing the median of both halves of a sliding window, repeated for 3 different windows lengths (320, 480 and 640 days for blue, red and yellow, respectively).**

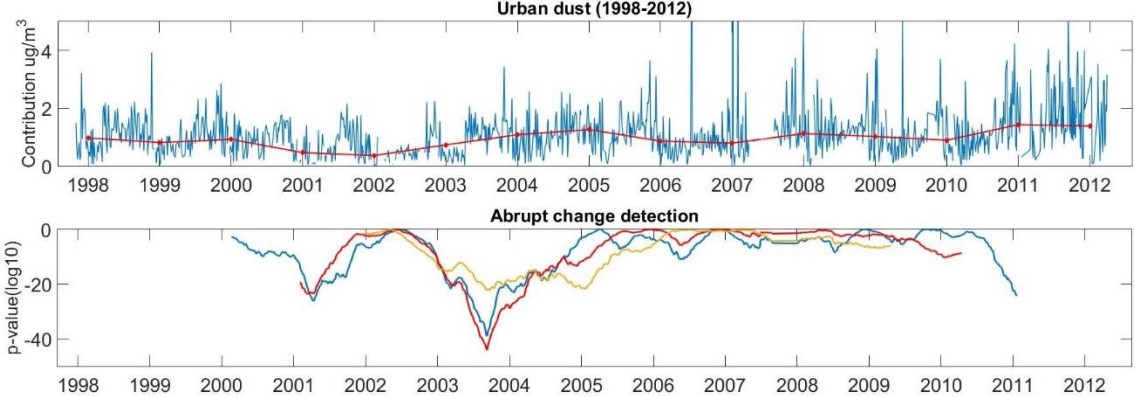

**Figure 9 Top panel: Time series of urban dust contribution to PM2.5 and the annual median in red. Bottom panel: p-value from a Mann-Whitney hypothesis test comparing the median of both halves of a sliding window, repeated for 3 different windows lengths (320, 480 and 640 days for blue, red and yellow, respectively).**

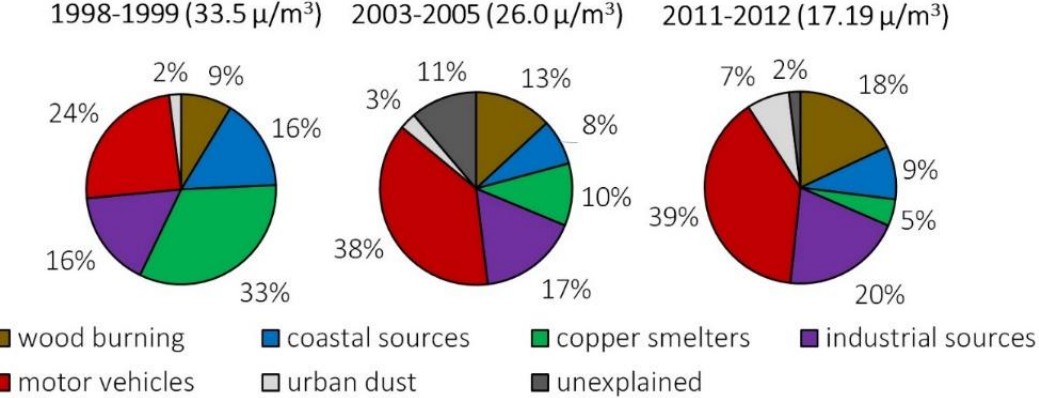

**Figure 10 Relative contribution of each source at the beginning, middle, and end of the period investigated in this study. Median levels of total PM2.5 are given in brackets next to the corresponding time period.**

**Table 1 Comparison of source apportionment studies in Latin American cities[a]. Total PM$_{2.5}$ and its sources are expressed in µg/m³.**

| Site Location | Country | Population | Model used | Reference | Study year | PM$_{2.5}$ | Sea salt | Dust | Traffic | Industry | Biomass burning | Other |
|---|---|---|---|---|---|---|---|---|---|---|---|---|
| Cordoba | Argentina | 1,272,000 | PMF | Lopez[66] | 2009/2010 | 71 | | 39.1 | 22.7 | 9.2 | | 0.0 |
| Curitiba | Brazil | 2,751,907 | APCA | Andrade[1] | 2007/2008 | 12 | 0.0 | 0.0 | 6.6 | 1.9 | 0.0 | 3.5 |
| Porto Alegre | Brazil | 1,409,351 | APCA | Andrade[1] | 2007/2008 | 16 | 0.0 | 0.0 | 5.6 | 0.5 | 0.0 | 9.9 |
| Belo Horizonte | Brazil | 2,375,151 | APCA | Andrade[1] | 2007/2008 | 17 | 0.0 | 7.5 | 3.1 | 2.0 | 0.0 | 4.4 |
| Recife | Brazil | 1,537,704 | APCA | Andrade[1] | 2007/2008 | 18 | 4.3 | 1.4 | 6.7 | | | 5.6 |
| Rio de Jainero | Brazil | 6,320,000 | APCA | Andrade[1] | 2007/2008 | 20 | | 2.8 | 10.2 | 3.6 | | 3.4 |
| Sao Paulo | Brazil | 11,235,503 | APCA | Andrade[1] | 2007/2008 | 28 | | 3.6 | 11.2 | 3.6 | | 9.5 |
| Rio de Janeiro | Brazil | 6,320,000 | APFA | Godoy[5] | 2003/2005 | 10 | | 3.5 | 2.8 | 3.4 | | 0.0 |
| Santiago | Chile | 5,278,000 | PMF | Jorquera[63–65] | 2004 | 32 | 3.2 | 1.3 | 10.0 | 3.1 | 9.3 | 5.3 |
| Santiago | Chile | 6,000,000 | CMB | Villalobos | 2013 | 33 | 1.0 | 2.5 | 11.0 | 4.6 | 5.2 | 8.9 |
| Moravia | Costa Rica | 56,919 | PMF | Murillo[67] | 2010/2011 | 18 | 2.0 | 3.9 | 5.2 | | | 6.9 |
| San Jose | Costa Rica | 288,054 | PMF | Murillo[67] | 2010/2011 | 26 | 2.0 | 3.5 | 4.8 | 6.9 | | 8.9 |
| Heredia | Costa Rica | 20,191 | PMF | Murillo[67] | 2010/2011 | 37 | 2.4 | 5.1 | 5.8 | 10.3 | | 13.4 |
| Tijuana | Mexico | 1,301,000 | PMF | Minguillon[50] | 2010 | 19 | 2.9 | | 2.6 | 0.4 | 7.1 | 5.6 |
| Mexico City | Mexico | 8,851,000 | CMB | Mugica[51] | 2006 | 50 | | 13.3 | 21.0 | 5.0 | | 10.7 |
| Salamanca | Mexico | 152,048 | PMF | Murillo[52] | 2006/2007 | 45 | | 7.3 | 5.8 | 8.2 | | 23.7 |

(a) Adapted from WHO, http://www.who.int/quantifying_ehimpacts/global/source_apport/en/.

**Table 2 Comparison of long-term source apportionment studies carried out in urban areas.**

| Location, period | PM2.5 mass | Motor vehicles | Sulfates + nitrates + ammonia | Biomass burning | Soil | Industry [c] |
|---|---|---|---|---|---|---|
| Los Angeles, CA, US, 2002-2013 | 17.5 | 3.3 | 9.6 | 1.1 | 1.0 | - |
| Rubidoux, CA, US, 2002-2013 | 19.5 | 3.7 | 12.2 | 0.8 | 0.9 | 0.1 |
| Detroit, US, 2001-2014 | 11.8 | 2.5 | 5.1 | 0.8 | 0.9 | 0.5 |
| Chicago, US, 2006-2014 | 10.3 | 2.2 | 4.8 | 0.9 | 0.4 | 1.1 |
| Sidney, Australia, 1998-2009 | 9.3 | 2.1 | 1.8 [a] | 2.7 | 0.3 | - |
| Hanoi, Vietnam, 2001-2008 | 54.0 | 21.6 | 15.7 [a] | 7.0 | 1.8 | 10.3 |
| Kuala Lumpur, Malaysia, 2002-2011 | 25.1 | 8.9 | 12.1 [b] | 2.3 | 0.8 | 12.0 |

(a) Only ammonium sulfate is reported.
(b) Sulfate was expressed as ammonium sulfate.
(c) Whenever more than one type of industrial source has been resolved, they have been lumped together in a single industrial category.

**Table 3 Seasonal PM2.5 and source contribution identified by a stratified regression of the contributions obtained by PMF 5.0. The concentration values are given in µg/m3 for each season and source, with corresponding standard errors at 95% confidence level within the brackets. The 24-hour Chilean standard for PM2.5 is 50 µg/m3 and the WHO guidelines is 25 µg/m3.**

| Source | Autumn | Winter | Spring | Summer |
|---|---|---|---|---|
| $PM_{2.5}$ | 43.9 (±19.3) | 48.8 (±18.8) | 16.0 (4.0) | 16.7 (±4.7) |
| Wood burning | 5.27 (±0.82) | 14.95 (±1.77) | 3.94 ± 0.48 | 2.85 ± 0.24 |
| coastal sources | 3.21 (±0.43) | 1.86 (±0.44) | 1.12 ± 0.42 | not significant |
| Copper Smelter | 5.67 (±0.62) | 3.62±(0.52) | 3.57 ± 0.24 | 3.20 ± 0.84 |
| Industrial Sources | 7.89 (±0.88) | 5.39 ±(1.04) | 6.10 ± 0.37 | 5.28 ± 0.59 |
| Vehicles | 11.70 (±0.74) | 10.84 ±(0.83) | 7.85 ± 0.64 | 7.72 ± 1.34 |
| Urban Dust | 2.57 (±0.60) | not significant | not significant | 2.34 ± 0.66 |
| Days over Chilean standard | 138 | 149 | 2 | 4 |
| Days over WHO guidelines | 265 | 257 | 32 | 45 |
| No of daily Samples | 343 | 315 | 292 | 294 |

**Table 4 Significant changes in PM2.5 sources in context with Santiago air quality improvement measures taken at that time.**

| Source | Date Change event | Impact over source contributions | Explanation and comments |
|---|---|---|---|
| Motor vehicles | 2002 | Reduction of 1.53 µg/m³ (15.59%) | Fuel quality improvement. Lead removed from gasoline |
| Motor vehicles | 2005 and 2006 | Increase of 1.60 µg/m³ (18.92%). | Operation of urban highways. |
| Motor vehicles | 2007 | Increase of 1.34 µg/m³ (13.42%). | Increase in number of private motorized vehicles due to poor implementation of Transantiago |
| Motor vehicles | late 2008-2010 | Reduction of 5.69 µg/m³ (43.97%) | Improvement to Transantiago |
| Industrial sources | 2002 | Reduction of 2.52 µg/m³ (34.33%) | Diesel fuel sulfur content reduction in 2001. |
| Industrial sources | 2005-2007 | Increase of 1.86 µg/m³ (45.04%). | Argentinean natural gas import reduction |
| Industrial sources | 2009-2010 | Reduction of 1.76 µg/m³ (31.17%) | Opening of Quintero terminal for LNG import through marine port |
| Cooper smelter | 1998-2002 | Reduction of 4.13 µg/m³ (69.04%) | Implementation of emission abatement technology in Caletones smelter |
| Cooper smelter | 2010-2011 | Reduction of 1.41 µg/m³ (64.66%) | Reduction of $SO_2$ and PM emissions in Caletones and Ventana smelters |
| Wood burning | 2007-2008 | Increase of 1.16 µg/m³ (43.39%). | Unknow |
| Wood burning | 2009-2010 | Reduction of 0.55 µg/m³ (16.98%) | Unknow |
| coastal sources | 2002-2005 | Reduction of 1.62 µg/m³ (77.46%) | Diesel sulfur content reduction |

| coastal sources | Since 2010 | Reduction of 1.05 µg/m$^3$ (76.17%) | Opening of Quintero terminal for LNG import through marine port |
|---|---|---|---|
| Urban dust | 2001-2002 | Reduction of 0.42 µg/m$^3$ (48.84%) | Lead-free gasoline introduction |
| Urban dust | Since 2004 | Increase of 0.67 µg/m$^3$ (171.78%). | Increase in the number of motorized car (annual growth rate of 4%) |
| Urban dust | Since 2011 | Increase of 0.48 µg/m$^3$ (51.61%). | Increase in the number of motorized car (annual growth rate of 7%), extended drought since 2010 |

**Supplementary material**

**Table S 1 Summary of previous Santiago source apportionment studies (each column shows percentage contribution to PM2.5).**

| Reference | (Rojas et al., 1990) | (Artaxo, 1996) | (Artaxo, 1998) | (Artaxo, 1999) | (Artaxo, 1999) | (Moreno et al., 2010) | (Jorquera and Barraza, 2012) | (Jorquera and Barraza, 2012) | (Villalobos et al., 2015) |
|---|---|---|---|---|---|---|---|---|---|
| Location in Santiago | Downtown | Downtown | Downtown | Downtown | East | Downtown | Las Condes | Las Condes | San Joaquin |
| Time period considered | January-February 1987 | July-august 1996 | July-august 1998 | June-December 1999 | June-December, 1999 | 1998-2007 | 1999 | 2004 | 2013 |
| Sulfates | 49 | | | | | 13.6 | 19 | 16 | |
| Sulfates + As | | | | 39 | 15 | | | | |
| Sulfates + copper smelters | | | 9.7 | | | | | | |
| Copper smelters | | 8.7 | | | | | 11 | 10 | |
| Sulfates + industry | | 64 | | | | | | | |
| Residual oil combustion + industry | | | 23.2 | | | | | | |
| Residual oil combustion | 13 | 1.9 | | | | 13.6 | | | |
| Motor vehicles + industry | | | | | 70 | | | | |
| Motor vehicles | | 16 | 35.8 | 40 | | 12.3 | 28 | 31 | |
| Wood burning | | | | | | | 25 | 29 | 19 |
| Wood burning + car exhausts | 5.6 | | | | | | | | |

| | | | | | | | | |
|---|---|---|---|---|---|---|---|---|
| Solid dust + wood burning | 26 | | | | | | | |
| Solid dust | | 15.5 | 31.3 | 17 | 7 | 24.6 | 4 | 4 |
| Solid dust + industry | 6.4 | | | | | | | |
| metallurgical | | | | 4 | | | | |
| Marine aerosol | | | | | | | 13 | 10 |
| Diesel emission | | | | | | | | | 8 |
| Gasoline vehicles | | | | | | | | | 9 |

**Figure S1 Example of replacement of missing data. Original data for Cl (in blue) with missing data filled using a custom-written algorithm (in red). Shown are a) Cl concentration on a logarithmic scale, b) zoom of a data range with small and large filled data gaps**

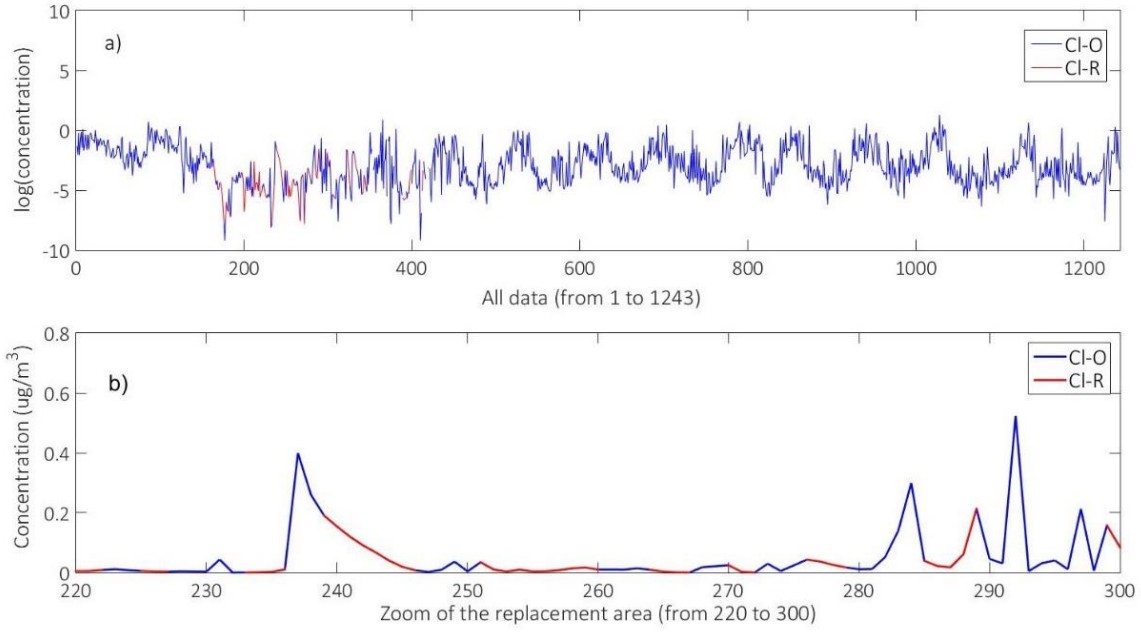

**Figure S2 K-Fe edge plot. To provide a tracer associated with wood burning we added the non-soil potassium parameter Kns calculated as Kns=K-0.3•Fe, from the K-Fe edge plot**

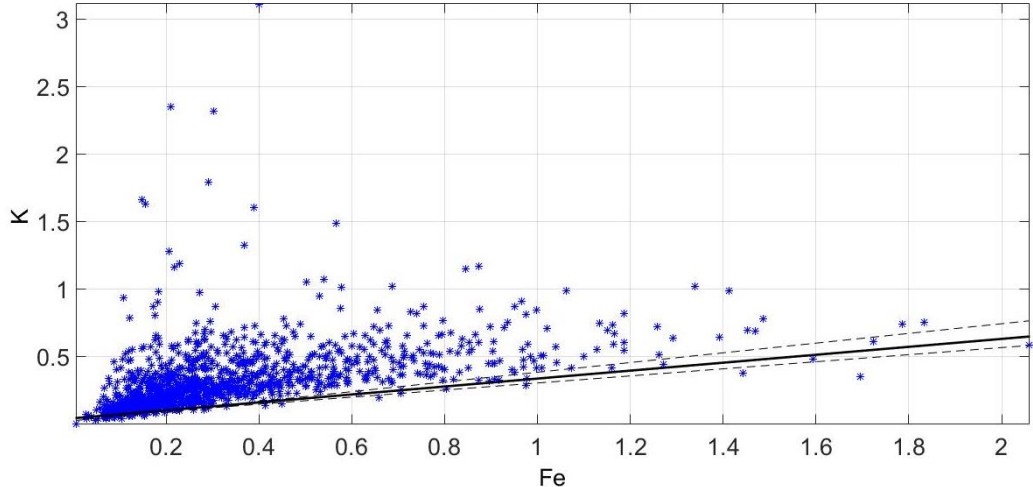

**Figure S3 Trend in Santiago motorized vehicles numbers (data provided by National institute of statistics, www.ine.cl) as well as sold vehicle fuel (data provided by Superintendence of electricity and fuels, www.sec.cl).**

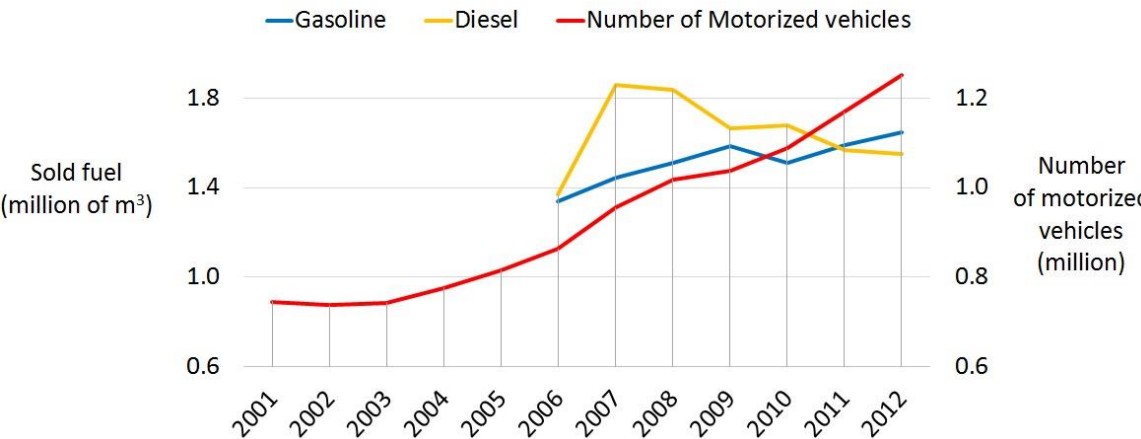

**Figure S4 Trend in Santiago vehicles annual growth rate (data provided by National institute of statistics, www.ine.cl) and contribution per vehicle to PM2.5 Santiago's levels. The contribution per vehicle was calculated by the dividing annual median motor vehicles contribution (from PMF) by the number of motorized vehicles in each year.**

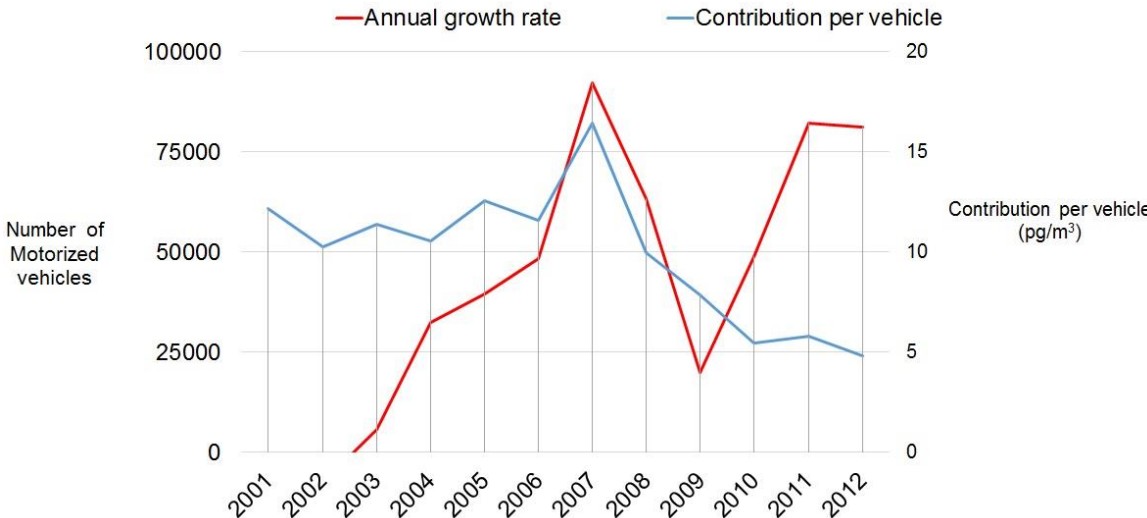

**Figure S5 Monthly precipitation anomalies from the mean in downtown Santiago, 1960-2015. Source: http://explorador.cr2.cl/**

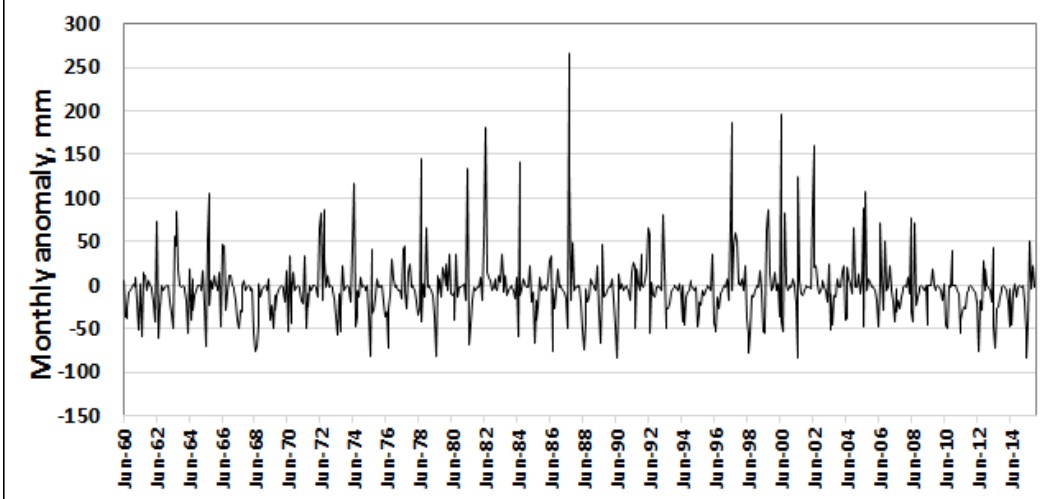