# Peer review of "Temporal evolution of main ambient PM2.5 sources in Santiago, Chile, from 1998 to 2012"

_Atmospheric Chemistry and Physics, 2017_

## Referee Comment (RC1) · Anonymous Referee #1 · 20 Feb 2017

Review of manuscript submitted to ACP- Atmos. Chem. Phys. Discuss., doi:10.5194/acp-2017-18, 2017

Temporal evolution of main ambient PM2.5 sources in Santiago, Chile, from 1998 to 2012 From: Francisco Barraza, Fabrice Lambert, Héctor Jorquera, Ana María Villalobos, Laura Gallardo.

Overall Assessment This study involved a very large number of samples and for a long time series: 1243 24-hour filter samples of ambient PM2.5 collected between April-1998 to August-2012. It was used two different source-receptor models (PMF 5.0 and Unmix 6.0). The detailed study shows that the main aerosol sources for PM2.5 were: motor vehicles (37%), industrial sources (19%), copper smelters (14%), wood burning (12%), coastal sources (10%), and urban dust (3%). For a very dry region, it is

surprising that urban dust is only 3% of aerosol mass, even considering that the analysis is for PM2.5. Some of the dust factor must have gone in the vehicular source of other factors. After analyzing the 15 years time series, the results show that over the 15 years, the emissions from motor vehicles, industrial sources, copper smelters, and coastal sources declined by about 21, 39, 81, 59, and 59% respectively, while wood burning didn't change and urban dust increase by 72%. Do you have an estimate for the standard deviation of this important result? The significance of these values depends on the standard deviations that are not reported. Are these reduction numbers all statistically significant at the 95% confidence interval? Another point is that it is not correct you say that the EMISSIONS were reduced, because you have not measure the emissions, but atmospheric concentrations. I think the best term would be: "The reduction of the impact of the different sources to atmospheric concentrations were". Also important is that there is a lack overall in the whole manuscript of standard deviation for the reported values. Even mean concentrations for PM2.5 do not report their standard deviation. The standard deviation is as important as the average value. I feel that in the overall manuscript and also in the reference list, there are very few references to similar studies in other cities. It looks as the study has no connections to other urban areas in Latin America and other places. It looks too isolated in the context on urban aerosol source apportionment. It is important to set the manuscript in a broader context of similar studies done in other urban areas, such as Mexico City, Sao Paulo, La Paz, Quito, etc, as well as some Indian cities that could share similar sources. There is an excess of Chilean studies reported, and a lack of other studies worldwide. Figure 2 shows that PMF has not separated residual oil combustion that UNIMIX attributes 7%. There is no discussion on why the two models provided such different results. Of course residual oil combustion must be present in Santiago. In PMF, where Vanadium and Nickel was attributed? This is an important issue that was not discussed in the manuscript.

Figures 3 to 9 shows boxplots that are difficult to read, and provide limited information with the outliners. I suggest only shows 50, 75 and 25 percentile, and forget about the

outliers, to improve the readability of the figures.

You discussed the impact of sources to PM2.5. What about the meteorology? Did it rain less? more? Cloud cover has changed? Wind direction has changed? Inversions got stronger? Since aerosol concentrations are a function of sources and meteorology, you need to discuss the possible changes in meteorology in detail.

I think that the study needs important improvements before it could be considered for publication in ACP. There are several important specific comments that needs to be addressed as well as the general comments discussed above.

Specific comments

Page 1 – line 18: the word WERE was missing Page 2 line 4 – instead of "impeding horizontal air movements", maybe it is better making it difficult the air mass transport over the metropolitan region. Page 2 Line 7 – It would be great to have more information on mixing layer heights than only the expression: The mixing layer shows a marked diurnal cycle (Saide et al., 2011)."". How much is the mixing layer height over winter and summer at midday? Frequency of thermal inversions? Etc... Page 3 line 5 – upperscript for ug/m$^3$... Page 4 – Line 33: The detection limit is an important variable, because of the phrase: "We decided to keep only those elements for which more than 70% of the samples contained valid measurements above the detection limit.". There are many ways to derive detection limits in XRF analysis. How were these derived? Blank variability was included? It was derived using 3 sigma? It was derived using multiple analysis of the same filter? How much was the average detection limit in ng/m$^3$ for each element?.

Page 4 – Which filter were used? I guess were 37 mm Teflon filters, but this needs to be explicitly mentioned. The same filter was used over the 12 years of sampling?

Page 4 The XRF methodology is described under the sampling station section, which is not correct. Suggestion: Open a new section to describe the XRF methodology.

Page 5 – Treatment of missing values: Again: This is described under the section sampling, and this is not appropriate. The treatment of missing values (up to 30% of the samples) is important and needs better description. Substitute the missing values by the median is certainly not appropriated, and I am surprised to see that the results using this wrong procedure and interpolation using better algorithms provide similar results. I really do not believe that this is the case. As up to 30% of data for some variables was artificially introduced in the analysis, a much better discussion on the effects must be provided in the manuscript. Page 5 Line 17: You need to define very precisely what you mean by BEST Model in the phrase: "We considered 13 species that yield the best model: Al, Si, S, Cl, non-soil K (Kns), Ti, Cr, Mn, Fe, Ni, Cu, Zn and As".

Page 5 lines 25-27. The factor 0.3 relating K to Fe as in the methodology of Lewis et al., can change a lot from site to site. You mentioned that you have done a regression, but you certainly needs much better explanation. You added a new variable that is not statistically independent from the others. This can bring problems in multivariate models. This needs to be much better discussed and explained.

Page 6 line 3: Even with a very high number of samples, you have NOT explained the origin of 25% of the variance of the PM2.5. Why you only explained 75% of the variance? This is a low value for multivariate analysis from urban areas. This is mentioned as " PMF 5.0 produces a six factors solution that explained 74% of the variance in ambient PM2.5 " You have not discussed this important point. It is strange that the unexplained PM2.5 is only 5-7% in Figure 2, and you explained only 75% of the variability.

Page 7 – Line 13: There is a very important lack overall in the whole manuscript of standard deviation for the reported values. Even mean concentrations for PM2.5 do not report their standard deviation. This is unacceptable in science: All reported average values needs to have their standard deviation reported together with the mean value. For instance in the phrase: "Over the whole study period, the daily mean (24 h)

concentration of PM2.5 was 35.60 $\mu$g/m3 and the median 24.19 $\mu$g/m3"

Page 8 line 20: Very strange that the reduction in industrial sources was attributed to decrease in sulfur in the DIESEL, used in the transportation. This needs to be correctly explained. This is on the phrase: "In Figure 5 we show temporal evolution of the source identify by PMF as industrial sources. This source reduced its 20 contributions from 1998 to 2012 by 2.63 $\mu$g/m3 (39.23%, p=0.11x10-8). This improvement can be explained by the reduction policies for sulfur in diesel fuel".
* * *

---

## Referee Comment (RC2) · Anonymous Referee #2 · 14 Mar 2017

"Temporal evolution of main ambient PM2.5 sources in Santiago Chile, from 1998 to 2012" General: The project seems to be carefully thought out. The analytical methodology (PMF 5.0 and Unmix 6.0) seems appropriate; however, a separate detailed sampling and QA/QC section is needed. Language and spellings need to be improved. Concentrations should be expressed in 3 significant figures throughout the text and in the figures and tables. The author should compare the data with other studies in urban areas. As such I recommend that it be published with major revision:

1) Page 3: "$\mu$g/m3" should be "$\mu$g/m3" - be consistent throughout the text, figures, and tables.

2) Page 3: "24-hour" or "24 hours" or 24 h" – be consistent with one of them.

3) Page 3: No mention for the sampling and analysis for PM2.5? How PM2.5 samples

were obtained? Which type of filter was used? Were the filters weighed in the clean room? Which analytical balance was used? Any QA/QC?

4) Page 3: A detailed QA/QC section for XRF analysis should be included. How often were the "QC" samples run? (What % age?). No estimates of recovery. What is the limit of quantitation? What is the uncertainty? Any blank correction? Precision and accuracy?

5) Page 4: Did the authors find selenium?

6) Page 4: Did the authors do the PMF analysis for the missing data? How was this handled?

7) Page 5: The contribution of Pb from industrial emissions cannot be ruled out. Motor vehicle is not the only source of Pb.

8) Page 7: "artefact" should be "artifact"

9) Page 8: "Gramsch et al. (2013)". Missing in the reference section.

10) Page 8 Lines 10 – 12: Did the private cars use diesel as a fuel? Primary source of BC are emissions from diesel engines, cook stoves, wood burning and forest fires.

11) Page 12: "Boisier, J.P., . . .. ..Mu?????oz, F.," should be corrected.
* * *

---

## Author Comment (AC1) · 15 May 2017

General comments

The authors give thanks to the reviewer for all the comments and questions that made it possible to vastly improve the manuscripts and the robustness of the results. The main changes to the new manuscript are focused in three areas: i) an appropriate description of the sampling and the quality assurance/quality control of the chemical data, ii) new comparative discussion of our results with other Latin American studies, but mostly with other cities abroad with similar multi-year data and methodology to ours, iii) more discussion about meteorological and climatic phenomena that play an important role on Santiago's air quality. More detail about the changes and answers to the reviewer are described below

Response to comments by Reviewer 1

1) Overall Assessment This study involved a very large number of samples and for a long time series: 1243 24-hour filter samples of ambient PM2.5 collected between April-1998 to August-2012. It was used two different source-receptor models (PMF 5.0 and Unmix 6.0). The detailed study shows that the main aerosol sources for PM2.5 were: motor vehicles (37%), industrial sources (19%), copper smelters (14%), wood burning (12%), coastal sources (10%), and urban dust (3%).

For a very dry region, it is surprising that urban dust is only 3% of aerosol mass, even considering that the analysis is for PM2.5. Some of the dust factor must have gone in the vehicular source of other factors.

Answer and comment:

The reviewer has pointed out that our estimated urban dust is low for an arid region. However, Santiago is rather a semi-arid zone, for 1998-2012, the annual precipitation had a mean of 320 mm, a low of 110 mm (in 1998) and a high of 620 mm (in 2002). If we compare our concentrations of PMF-resolved soil factor in Santiago (average: 1.1 $\mu$g/m3) with those estimated in several cities in California, USA (a region with a very similar climate) we find that: a) Schauer et al (1996) have reported urban dust between 0.5 and 0.9 $\mu$g/m3 (6.8 – 14.3%) in Pasadena, Dowtown LA, West LA and Rubidoux, CA, as an annual average for 1982, using CMB as receptor model. b) Wang and Hopke (2013) have reported a 10-year source apportionment (PMF) at San Jose, CA and have found an average road dust of 0.58 $\mu$g/m3 (5.1%) for 2002-2012 c) Hasheminassab et al (2014) have analyzed ambient PM2.5 at Central Los Angeles and Rubidoux, CA for the period 2002-2013. They have found, using PMF, an average soil contribution of 0.8 – 1.1 $\mu$g/m3 at both sites (5 and 6%, respectively). d) Kim et al (2010) have analyzed data between 2003 and 2005 at the two sites above, using PMF. They have found that the soil contribution varies between 1.5 and 2.0 $\mu$g/m3 (6.9 and 9.8%)for Central Los Angeles and 1.6-1.9 $\mu$g/m3 (6.0-7.6%) for Rubidoux, CA.

Furthermore, in other long term source apportionment studies carried out using PMF, the PMF resolved soil contribution (in $\mu$g/m3) is similar in magnitude: 0.6 for the Sidney Basin between 1998 – 2009 (Cohen et al, 2011), 1.6 for Hanoi, Vietnam between 2001-2008 (Cohen et al, 2010), 0.5 and 0.8 in Detroit and Chicago for 2001-2014 (Milando et al, 2016).

Therefore, our results, on a mass basis, are within the values reported at urban sites with a similar Mediterranean climate as well as in other cities. Nonetheless, we agree with the reviewer that some urban dust may be mixed in with the motor vehicle source.

In the revised manuscript, we have added section 1.1 and two tables (1 and 2):

1.1 Source apportionment data analyses

Receptor models (see below) are state-of-the-art computational tools that allow researchers to identify and quantify the major sources that contribute to ambient PM2.5 concentrations in a given region and over a given period. Within the Latin American region, several source apportionment studies have been carried out in the largest cities such as Mexico City (Mugica et al., 2002), Sao Paulo, Brazil (Andrade et al., 2012), Rio de Janeiro, Brazil (Andrade et al., 2012; Godoy et al., 2009), and Santiago, Chile (Jorquera and Barraza, 2012; Villalobos et al., 2015). However, all these studies spanned only 1 - 2 years, were carried out using different receptor models, and differed in the time period analyzed, so it is difficult to quantitatively compare among them. Nonetheless, traffic and industrial sources are the typical major contributors to ambient PM2.5 as shown in Table 1, while biomass burning is relevant only in some cities. The 'other' category source is relevant in most Latin American cities and it may be due to processes leading to organic and inorganic PM2.5, plus smaller unresolved sources such as meat cooking, combustion of natural gas, coal, liquefied petroleum gas, etc. (WHO, 2017). Although these studies provide a quantitative assessment of ambient PM2.5 sources, we are aware of no long-term urban source apportionment studies in Latin America. Long-term studies provide a quantitative estimation of the temporal evolution

of major contributing sources, so an evaluation of the effectiveness of sector regulations can be performed. This information is critical for policy-makers and stakeholders, to provide feedback and suggest new initiatives to further reduce pollution levels. Table 2 below summarizes several long-term studies carried out in developed and developing countries within a similar period. Motor vehicles and industrial source contributions are clearly higher in developing countries (including most Latin American cities - Table 1), whereas in developed countries those sources have been controlled and their contributions are lower. See attached table 1 and 2

2) After analyzing the 15 years time series, the results show that over the 15 years, the emissions from motor vehicles, industrial sources, copper smelters, and coastal sources declined by about 21, 39, 81, 59, and 59% respectively, while wood burning didn't change and urban dust increase by 72%. Do you have an estimate for the standard deviation of this important result? The significance of these values depends on the standard deviations that are not reported. Are these reduction numbers all statistically significant at the 95% confidence interval? Another point is that it is not correct you say that the EMISSIONS were reduced, because you have not measure the emissions, but atmospheric concentrations. I think the best term would be: "The reduction of the impact of the different sources to atmospheric concentrations were". Also important is that there is a lack overall in the whole manuscript of standard deviation for the reported values. Even mean concentrations for PM2.5 do not report their standard deviation. The standard deviation is as important as the average value.

Answer and comment:

We certainly agree with the reviewer on this, so now we quote uncertainties for all reported results. Likewise, the trends that come from robust linear regression of model source contributions against time are now reported with their 95% confidence intervals; in this way, we can conclude about the significance of each one. On the other hand, the standard deviation is not the best parameter to describe dispersion of non-normally distributed data, so we have used the MAD (median absolute deviations) in the revised

manuscript. In discussing our results, we now refer to 'impacts' or contributions rather than to 'emissions'.

For example, the following paragraph is from the abstract in the revised manuscript:

PMF resolved six sources that contributed to ambient PM2.5, with UNMIX producing similar results: motor vehicles (37.3±1.1%), industrial sources (18.5±1.3%), copper smelters (14.4±0.8%), wood burning (12.3±1.0%), coastal sources (9.5±0.7%), and urban dust (3.0±1.2%). Our results show that over the 15 years analyzed here, four of the resolved sources significantly decreased [95% Confidence Interval]: motor vehicles 21.3% [2.6, 36.5], industrial sources 39.3% [28.6, 48.4], copper smelters 81.5% [75.5, 85.9], and coastal sources 58.9% [38.5, 72.5], while wood burning didn't significantly change, and urban dust increased by 72% [48.9, 99.9].

3) I feel that in the overall manuscript and also in the reference list, there are very few references to similar studies in other cities. It looks as the study has no connections to other urban areas in Latin America and other places. It looks too isolated in the context on urban aerosol source apportionment. It is important to set the manuscript in a broader context of similar studies done in other urban areas, such as Mexico City, Sao Paulo, La Paz, Quito, etc, as well as some Indian cities that could share similar sources. There is an excess of Chilean studies reported, and a lack of other studies worldwide.

Answer and comment:

We agree with the reviewer. We have now included a revised Table 1 that summarizes source apportionment studies conducted in Latin American cities (WHO, 2017). We have commented on the similarities among them in the introductory section.

We did not find any long-term source apportionment studies in Latin American cities. Therefore, we have summarized long-term source apportionment studies carried out abroad, with an emphasis on California because of the similarities with Santiago's climate. We have added a new Table 2 with comparisons with the following long-term PM2.5 source apportionment studies (all carried out using PMF) that we have found in the literature:

a) 2002-2012 in San Jose, CA (Wang and Hopke, 2013) b) 2002-2013 in Central Los Angeles and Rubidoux, CA (Hasheminassab et al., 2014) c) 2002-2011 in Detroit and Chicago (Milando et al., 2016) d) 1998-2009 in Sidney, Australia (Cohen et al., 2011) e) 2002-2011 in Kuala Lumpur, Malasya (Rahman et al., 2015) f) 2001-2008 in Hanoi, Vietnam (Cohen et al., 2010)

We have also commented upon these studies in the introduction section (see answer to question 1 above).

4) Figure 2 shows that PMF has not separated residual oil combustion that UNIMIX attributes 7%. There is no discussion on why the two models provided such different results. Of course residual oil combustion must be present in Santiago. In PMF, where Vanadium and Nickel was attributed? This is an important issue that was not discussed in the manuscript.

Answer and comment:

UNMIX resolved the oil combustion as a unique source, apportioning 85% of Ni concentration to that source profile. The PMF solution apportions 56.5 and 19.5% of Ni concentration to the motor vehicles and industrial source profiles, respectively; in other words, PMF mixes sources that come together at the receptor site, transported by winds. This is a consequence of the different methodologies used by PMF and UNMIX to compute source profiles.

We acknowledge that vanadium and nickel are good tracers for oil industrial combustion, but we removed vanadium from the model, because we couldn't obtain a PMF solution using this element. This was due to prolonged periods with vanadium values below LOD between 2002 and 2006. During that period, Santiago's industry used

natural gas as industrial fuel, explaining those low vanadium records.

We have added the above two paragraphs in the discussion section.

5) Figures 3 to 9 shows boxplots that are difficult to read, and provide limited information with the outliners. I suggest only shows 50, 75 and 25 percentile, and forget about the outliers, to improve the readability of the figures.

Answer and comment:

We agree with the reviewer. We have improved those figures. As an example, we show below two of those figures

See attached figures 3 and 4 from the revised manuscript.

6) You discussed the impact of sources to PM2.5. What about the meteorology? Did it rain less? more? Cloud cover has changed? Wind direction has changed? Inversions got stronger? Since aerosol concentrations are a function of sources and meteorology, you need to discuss the possible changes in meteorology in detail. I think that the study needs important improvements before it could be considered for publication in ACP. There are several important specific comments that needs to be addressed as well as the general comments discussed above.

Answer and comment:

We do agree, and we have completed the discussion on meteorology. In fact, Central Chile is characterized by significant inter-annual variability in connection with El Niño Southern Oscillation (ENSO) (Garreaud et al., 2009) This characteristic inter-annual variability is illustrated in Supplementary figure 5 that shows monthly anomalies in precipitation (mm) registered in Santiago downtown since 1960 (Data available at http://explorador.cr2.cl/).

See Figure S5

We can see from figure S5 that there was a downward trend in annual precipitation between 1998 and 2012. Furthermore, since 2007, central and southern Chile has been affected by an extended and persistent drought, partly caused by natural variability and partly linked to a global warming trend ((CR2), 2015; Boisier et al., 2016). We think this drought is a contributing factor in explaining the increase in soil dust contribution in our PMF solution. Likewise, the above downward trend in precipitation implies a worsening of ventilation in Santiago's basin along the period analyzed. However, our trend analysis shows four major PM2.5 sources decreasing their contributions in the same period. The fact that all our trend estimates for those four sources were negative and significant means that they are conservative estimates, because we did not adjust them for meteorological conditions. The latter computation is beyond the scope of this study because it would require inverse modeling to estimate source strengths.

Regarding mixing height observations, the Chilean Meteorological Service does not launch radiosondes in Santiago, except for limited, short-term campaigns. The only data available are collected with a ceilometer since 2008 (Muñoz et al., 2010; Muñoz and Alcafuz, 2012); these data have significant diurnal, day-to-day, seasonal and possibly inter-annual variability. The following figure illustrates boundary layer height (BLH) retrieved at the Geophysics Department in downtown Santiago between January 1st, 2007 and December 31st 2013 (data kindly provided by Prof. Muñoz). The methodology for the retrievals is described in Muñoz and Undurraga (2010), and considers cloud-free data between 10 and 15 local time (UTC-4). Figure A6 shows a clear seasonality in BLH, with peak values in the austral summer season and lowest values during the austral winter. We see no apparent temporal trend on BLH, so we think this meteorological variable played no role in the temporal trends estimated for all sources resolved by the receptor model analysis.

Figure A6

We have added these new comments to the introduction and discussion sections.

7) Specific comments

Page 1 – line 18: the word WERE was missing

Answer: This has been corrected in the revised manuscript Page 2 line 4 – instead of "impeding horizontal air movements", maybe it is better making it difficult the air mass transport over the metropolitan region.

Answer: This has been corrected in the revised manuscript Page 2 Line 7 – It would be great to have more information on mixing layer heights than only the expression: The mixing layer shows a marked diurnal cycle (Saide et al., 2011)."". How much is the mixing layer height over winter and summer at midday? Frequency of thermal inversions? Etc. . .

Answer: This has been corrected in the revised manuscript. Please see the above response regarding meteorological variables in Santiago.

Page 3 line 5 –upperscript for ug/m3.

Answer: This has been corrected in the revised manuscript Page 4 – Line 33: The detection limit is an important variable, because of the phrase: "We decided to keep only those elements for which more than 70% of the samples contained valid measurements above the detection limit.". There are many ways to derive detection limits in XRF analysis. How were these derived? Blank variability was included? It was derived using 3 sigma? It was derived using multiple analysis of the same filter? How much was the average detection limit in ng/m3 for each element?.

Answer: A new section on Laboratory and QA/QC analysis was added with this information. For each species LOD was calculated as 3 times the standard deviation of field blanks. Page 4 – Which filter were used? I guess were 37 mm Teflon filters, but this needs to be explicitly mentioned. The same filter was used over the 12 years of sampling?

Answer: A new section on Laboratory and QA/QC analysis was added with this information. 37 mm Teflon filter were used and the methodology was the same throughout

the 15 years.

Page 4 The XRF methodology is described under the sampling station section, which is not correct. Suggestion: Open a new section to describe the XRF methodology.

Answer: A new section on Laboratory and QA/QC analysis was added with this information.

Below we present the section on QA/QC included in the revised manuscript that answers the above three specific comments.

2.2 Laboratory and QA/QC analysis Filters were inspected before being used, and the particles' concentration were determined gravimetrically using a microbalance, with a resolution of 0.01 mg. All filter (blank and filter samples) were stored at constant temperature (22±3°C) and relative humidity (40% HR ±3%) for a least 24-hours before being weighed. Those filters were analyzed using X-ray fluorescence (XRF) at the Desert Research Institute, Reno, NV, USA. The Ministry for the Environment provided the database containing the elemental analyses of those filters. In order to build statistical models based on robust chemical signals, we decided to keep only those elements selected in other studies that used the same data (CMM-MMA, 2011; Koutrakis et al., 2005; Sax et al., 2007; Valdes et al., 2012), for which more than 75% of the samples contained valid measurements above the detection limit. The limit of detection (LOD) was calculated for each element as three times the standard deviation of the blank (blanks represented approximately 10% of the samples). This public database (gravimetry and elemental analysis) has been used in several studies and all of them have already described the laboratory and QA/QC methodology (CMM-MMA, 2011; Jhun et al., 2013; Koutrakis et al., 2005; Sax et al., 2007). Thus, out of the 49 elements reported, we only kept 22: Na, Mg, Al, Si, S, Cl, K, Ca, Ti, V, Cr, Mn, Fe, Ni, Cu, Zn, As, Se, Br, Sr, Ba and Pb.

Page 5 – Treatment of missing values: Again: This is described under the section sampling, and this is not appropriate. The treatment of missing values (up to 30% of

the samples) is important and needs better description. Substitute the missing values by the median is certainly not appropriated, and I am surprised to see that the results using this wrong procedure and interpolation using better algorithms provide similar results. I really do not believe that this is the case. As up to 30% of data for some variables was artificially introduced in the analysis, a much better discussion on the effects must be provided in the manuscript.

Answer: Indeed, the interpolation procedure provided better results. We have included a more detailed description of that algorithm in the revised manuscript, as follows:

The missing data in those twenty-two species were dealt with as follows. First, we let the receptor models' (PMF5, UNMIX6) internal algorithms deal with them, which consists of replacing missing values with the median of the complete time-series for each species. Since replacing missing data with the median can lead to severe distortions in the data, we have also used a custom-written algorithm. This method interpolates up to three consecutive missing values using MATLAB's piecewise cubic interpolation algorithm. Sections of four or more consecutive missing values are filled in by summing up a mirrored copy of equal data length on both sides of the missing records, weighted by a cos2 function, thus ensuring that no artificial frequencies or discontinuities are introduced in the signal - the data filling algorithm is a MATLAB code that is available upon request to F. Lambert. We ensured that only relatively small missing data gaps were filled by using this method, so no large data sections were artificially created. The original and interpolated data are shown in Supplementary figure S1. Since our custom algorithm does not introduce discontinuities in the time series, we used this method for our analysis. In contrast, the receptor model results using the median-based missing data replacement can lose the seasonal signal of some species. Accordingly, the model results using the median-based filling algorithm yielded more variability in Cl, Ti Cr, Ni and As (species with important number of blanks). See supplementary figure S1

Page 5 Line 17: You need to define very precisely what you mean by BEST Model in the phrase: "We considered 13 species that yield the best model: Al, Si, S, Cl, non-soil

K (Kns), Ti, Cr, Mn, Fe, Ni, Cu, Zn and As”.

Answer: The “best model” means the model that has the best regression parameters and at the same time the highest number of resolved sources (statistically significant regression coefficients). We have added the next sentence:

“This model had the most robust regression parameter and the highest number of statistically significant source factors”

Page 5 lines 25-27. The factor 0.3 relating K to Fe as in the methodology of Lewis et al., can change a lot from site to site. You mentioned that you have done a regression, but you certainly needs much better explanation.

Answer: Indeed, the factor 0.3 is specific for the sampling site and was calculated with the data collected therein. We have added a supplementary figure with the K-Fe scatter plot showing a lower edge with a 0.3 slope (see figure in our next answer below).

You added a new variable that is not statistically independent from the others. This can bring problems in multivariate models. This needs to be much better discussed and explained.

Answer: We knew about this problem, so we have added Kns in and removed K from the model. We have added the following sentence to the manuscript:

To provide a tracer associated with wood burning, we have added the non-soil potassium parameter Kns calculated as Kns=(K-0.3xFe) and removed K from the model. The 0.3 coefficient was obtained from a K-Fe edge plot (supplementary figure 2)”

See Supplementary Figure S2 Page 6 line 3: Even with a very high number of samples, you have NOT explained the origin of 25% of the variance of the PM2.5. Why you only explained 75% of the variance? This is a low value for multivariate analysis from urban areas. This is mentioned as ” PMF 5.0 produces a six factors solution that explained 74% of the variance in ambient PM2.5 “ You have not discussed this important point. It is strange that the unexplained PM2.5 is only 5-7% in Figure 2, and you explained only

75% of the variability.

Answer: The average source contributions must add up to the average PM2.5 concentration by design (except by an intercept value). On the other hand, model variability stands for the scatter of individual (daily) model estimates compared with the actual observed (daily) PM2.5. These are different parameters. It is difficult to find receptor models that explain more than 90% of the variance. In our case, the unexplained 25% may be ascribed to the following causes:

i) Actual source profiles do not stay constant over the whole modeling period, adding uncertainty to the model results. ii) We could not include organic/elemental carbon into the model. The lack of these two components may increase uncertainty in the resolved source profiles. For instance, the ratio OC/EC is helpful in discriminating motor vehicles from wood burning. iii) We only had inorganic tracer species in the dataset, which put a limitation on our analysis. We knew that secondary organic aerosols (SOA) may be relevant in the warm season (October – April), as shown by Villalobos et al (2015) for Santiago in 2013 (application of CMB with organic molecular markers); these authors estimated that nearly 30% of total PM2.5 was identified as SOA.

We have added these new comments in the discussion section.

Page 7 – Line 13: There is a very important lack overall in the whole manuscript of standard deviation for the reported values. Even mean concentrations for PM2.5 do not report their standard deviation. This is unacceptable in science: All reported average values needs to have their standard deviation reported together with the mean value. For instance in the phrase: "Over the whole study period, the daily mean (24 h) concentration of PM2.5 was 35.60 $\mu$g/m3 and the median 24.19 $\mu$g/m3"

Answer: We agree with this comment. Since data have a non-normal distribution, we have used the median absolute deviation (MAD) as a measure of dispersion for each reported median and mean value.

Page 8 line 20: Very strange that the reduction in industrial sources was attributed to decrease in sulfur in the DIESEL, used in the transportation. This needs to be correctly explained. This is on the phrase: “In Figure 5 we show temporal evolution of the source identify by PMF as industrial sources. This source reduced its 20 contributions from 1998 to 2012 by 2.63 $\mu$g/m3 (39.23%, p=0.11x10-8). This improvement can be explained by the reduction policies for sulfur in diesel fuel”.

Answer: The quality of industrial diesel fuel was also improved. We put in the page 8 (3.3.2 section) the following new sentence: “[. . .], that can be explained by a reduction of sulfur in industrial diesel, which was reduced from 1000 to 300 ppm in 2001 [. . .]”

New references

(CR2), C. for C. and R.R., 2015. Mega Drought (2010-2015): a lesson to the future. Report for policymakers (in Spanish).

Boisier, J.P., Rondanelli, R., Garreaud, R.D., Munoz, F., 2016. Anthropogenic and natural contributions to the Southeast Pacific precipitation decline and recent megadrought in central Chile. Geophys. Res. Lett. 43, 413–421. doi:10.1002/2015GL067265

Cohen, D.D., Crawford, J., Stelcer, E., Bac, V.T., 2010. Characterisation and source apportionment of fine particulate sources at Hanoi from 2001 to 2008. Atmos. Environ. 44, 320–328. doi:10.1016/j.atmosenv.2009.10.037

Cohen, D.D., Stelcer, E., Garton, D., Crawford, J., 2011. Fine particle characterisation, source apportionment and long-range dust transport into the Sydney Basin: a long term study between 1998 and 2009. Atmos. Pollut. Res. 2, 182–189. doi:10.5094/APR.2011.023

Garreaud, R.D., Vuille, M., Compagnucci, R., Marengo, J., 2009. Present-day South American climate. Palaeogeogr. Palaeoclimatol. Palaeoecol. 281, 180–195. doi:10.1016/j.palaeo.2007.10.032

Hasheminassab, S., Daher, N., Ostro, B.D., Sioutas, C., 2014. Long-term source apportionment of ambient fine particulate matter (PM 2.5) in the Los Angeles Basin: A focus on emissions reduction from vehicular sources. Environ. Pollut. 193, 54–64. doi:10.1016/j.envpol.2014.06.012

Lewis, C.W., Norris, G. a, Conner, T.L., Henry, R.C., 2003. Source apportionment of Phoenix PM2.5 aerosol with the Unmix receptor model. J. Air Waste Manag. Assoc. 53, 325–338. doi:10.1080/10473289.2003.10466155

Kim, E., Turkiewicz, K., Zulawnick, S.A., Magliano, K.L., 2010. Sources of fine particles in the South Coast area, California. Atmos. Environ. 44 (26), 3095-3100. doi: 10.1016/j.atmosenv.2010.05.037
* * *
Table 1 Comparison of source apportionment studies in Latin American cities[(a)]. Total $PM_{2.5}$ and its sources are expressed in μg/m³.

| Site Location | Country | Population | Model used | Reference | Study year | $PM_{2.5}$ | Sea salt | Dust | Traffic | Industry | Biomass burning | Other |
|---|---|---|---|---|---|---|---|---|---|---|---|---|
| Cordoba | Argentina | 1,272,000 | PMF | Lopez[66] | 2009/2010 | 71 | | 39.1 | 22.7 | 9.2 | | 0.0 |
| Curitiba | Brazil | 2,751,907 | APCA | Andrade[1] | 2007/2008 | 12 | 0.0 | 0.0 | 6.6 | 1.9 | 0.0 | 3.5 |
| Porto Alegre | Brazil | 1,409,351 | APCA | Andrade[1] | 2007/2008 | 16 | 0.0 | 0.0 | 5.6 | 0.5 | 0.0 | 9.9 |
| Belo Horizonte | Brazil | 2,375,151 | APCA | Andrade[1] | 2007/2008 | 17 | 0.0 | 7.5 | 3.1 | 2.0 | 0.0 | 4.4 |
| Recife | Brazil | 1,537,704 | APCA | Andrade[1] | 2007/2008 | 18 | 4.3 | 1.4 | 6.7 | | | 5.6 |
| Rio de Jainero | Brazil | 6,320,000 | APCA | Andrade[1] | 2007/2008 | 20 | | 2.8 | 10.2 | 3.6 | | 3.4 |
| Sao Paulo | Brazil | 11,235,503 | APCA | Andrade[1] | 2007/2008 | 28 | | 3.6 | 11.2 | 3.6 | | 9.5 |
| Rio de Janeiro | Brazil | 6,320,000 | APFA | Godoy[5] | 2003/2005 | 10 | | 3.5 | 2.8 | 3.4 | | 0.0 |
| Santiago | Chile | 5,278,000 | PMF | Jorquera[63–65] | 2004 | 32 | 3.2 | 1.3 | 10.0 | 3.1 | 9.3 | 5.3 |
| Santiago | Chile | 6,000,000 | CMB | Villalobos | 2013 | 33 | 1.0 | 2.5 | 11.0 | 4.6 | 5.2 | 8.9 |
| Moravia | Costa Rica | 56,919 | PMF | Murillo[67] | 2010/2011 | 18 | 2.0 | 3.9 | 5.2 | | | 6.9 |
| San Jose | Costa Rica | 288,054 | PMF | Murillo[67] | 2010/2011 | 26 | 2.0 | 3.5 | 4.8 | 6.9 | | 8.9 |
| Heredia | Costa Rica | 20,191 | PMF | Murillo[67] | 2010/2011 | 37 | 2.4 | 5.1 | 5.8 | 10.3 | | 13.4 |
| Tijuana | Mexico | 1,301,000 | PMF | Minguillon[50] | 2010 | 19 | 2.9 | | 2.6 | 0.4 | 7.1 | 5.6 |
| Mexico City | Mexico | 8,851,000 | CMB | Mugica[51] | 2006 | 50 | | 13.3 | 21.0 | 5.0 | | 10.7 |
| Salamanca | Mexico | 152,048 | PMF | Murillo[52] | 2006/2007 | 45 | | 7.3 | 5.8 | 8.2 | | 23.7 |

(a) Adapted from WHO, http://www.who.int/quantifying_ehimpacts/global/source_apport/en/.

**Fig. 1.**

Table 2 Comparison of long-term source apportionment studies carried out in urban areas.

| Location, period | PM2.5 mass | Motor vehicles | Sulfates + nitrates + ammonia | Biomass burning | Soil | Industry [c] |
|---|---|---|---|---|---|---|
| Los Angeles, CA, US, 2002-2013 | 17.5 | 3.3 | 9.6 | 1.1 | 1.0 | - |
| Rubidoux, CA, US, 2002-2013 | 19.5 | 3.7 | 12.2 | 0.8 | 0.9 | 0.1 |
| Detroit, US, 2001-2014 | 11.8 | 2.5 | 5.1 | 0.8 | 0.9 | 0.5 |
| Chicago, US, 2006-2014 | 10.3 | 2.2 | 4.8 | 0.9 | 0.4 | 1.1 |
| Sidney, Australia, 1998-2009 | 9.3 | 2.1 | 1.8 [a] | 2.7 | 0.3 | - |
| Hanoi, Vietnam, 2001-2008 | 54.0 | 21.6 | 15.7 [a] | 7.0 | 1.8 | 10.3 |
| Kuala Lumpur, Malaysia, 2002-2011 | 25.1 | 8.9 | 12.1 [b] | 2.3 | 0.8 | 12.0 |

(a) Only ammonium sulfate is reported.
(b) Sulfate was expressed as ammonium sulfate.
(c) Whenever more than one type of industrial source has been resolved, they have been lumped together in a single industrial category.

**Fig. 2.**

Figure 3 Temporal evolution of $PM_{2.5}$ concentrations in Parque O'Higgins monitoring station in central Santiago. The red line shows the annual median.

[Figure]

Figure 4 Top panel: Time series of motor vehicles contribution to $PM_{2.5}$. Bottom panel: p-value from a hypothesis test comparing the medians of both halves of a sliding window, repeated for 3 different windows lengths (320, 480 and 640 days for blue, red and yellow, respectively).

[Figure]

Fig. 3.

Figure S5 Monthly precipitation anomalies from the mean in downtown Santiago, 1960-2015. Source: http://explorador.cr2.cl/

[Figure]

**Fig. 4.**

ACPD
[Figure]

Fig. 5.
Figure S1 Replacement of missing data. Original data (red line) with missing data filled using a custom-written algorithm (blue line) for Cl, Ti, Cr, Ni, Cu and As are show on the left. The right hand side of the figure shows a zoom of the left panels for selected periods.

[Figure]

Fig. 6.

Figure S2 K-Fe edge plot. To provide a tracer associated with wood burning we added the non-soil potassium parameter Kns calculated as Kns=K-0.3•Fe, from the K-Fe edge plot

[Figure]

Fig. 7.

---

## Author Comment (AC2) · 15 May 2017

General comments

The authors give thanks to the reviewer for all the comments and questions that made it possible to vastly improve the manuscripts and the robustness of the results. The main changes to the new manuscript are focused in three areas: i) an appropriate description of the sampling and the quality assurance/quality control of the chemical data, ii) new comparative discussion of our results with other Latin American studies, but mostly with other cities abroad with similar multi-year data and methodology to ours, iii) more discussion about meteorological and climatic phenomena that play an important role on Santiago's air quality. More detail about the changes and answers to the reviewer are described below

Response to comments by Reviewer 2

General: The project seems to be carefully thought out. The analytical methodology (PMF 5.0 and Unmix 6.0) seems appropriate; however, a separate detailed sampling and QA/QC section is needed. Language and spellings need to be improved. Concentrations should be expressed in 3 significant figures throughout the text and in the figures and tables. The author should compare the data with other studies in urban areas. As such I recommend that it be published with major revision:

1) Page 3: "$\mu$g/m3" should be "$\mu$g/m3" - be consistent throughout the text, figures, and tables.

Answer: This was corrected in the revised manuscript

2) Page 3: "24-hour" or "24-hours" or 24 h" – be consistent with one of them.

Answer: This was corrected in the revised manuscript

3) Page 3: No mention for the sampling and analysis for PM2.5? How PM2.5 samples were obtained? Which type of filter was used? Were the filters weighed in the clean room? Which analytical balance was used? Any QA/QC?

Answer: the follow section was added with this information.

2.2 Laboratory and QA/QC analysis Filters were inspected before being used, and the particles' concentration were determined gravimetrically using a microbalance, with a resolution of 0.01 mg. All filter (blank and filter samples) were stored at constant temperature (22$\pm$3°C) and relative humidity (40% HR $\pm$3%) for a least 24-hours before being weighed. Those filters were analyzed using X-ray fluorescence (XRF) at the Desert Research Institute, Reno, NV, USA. The Ministry for the Environment provided the database containing the elemental analyses of those filters. In order to build statistical models based on robust chemical signals, we decided to keep only those elements selected in other studies that used the same data (CMM-MMA, 2011; Koutrakis et al., 2005; Sax et al., 2007; Valdes et al., 2012), for which more than 75% of the sam-

ples contained valid measurements above the detection limit. The limit of detection (LOD) was calculated for each element as three times the standard deviation of the blank (blanks represented approximately 10% of the samples). This public database (gravimetry and elemental analysis) has been used in several studies and all of them have already described the laboratory and QA/QC methodology (CMM-MMA, 2011; Jhun et al., 2013; Koutrakis et al., 2005; Sax et al., 2007). Thus, out of the 49 elements reported, we only kept 22: Na, Mg, Al, Si, S, Cl, K, Ca, Ti, V, Cr, Mn, Fe, Ni, Cu, Zn, As, Se, Br, Sr, Ba and Pb. The missing data in these twenty-two species were dealt with as follows. First, we let the receptor models' (PMF5, UNMIX6) internal algorithms deal with them, which consists of replacing missing values with the median of the complete time-series for each species. Since replacing missing data with the median can lead to severe distortions in the data, we have also used a custom-written algorithm. This method interpolates up to three consecutive missing values using MATLAB's piecewise cubic interpolation algorithm. Sections of four or more consecutive missing values are filled in by summing up a mirrored copy of equal data length on both sides of the missing records, weighted by a cos2 function, thus ensuring that no artificial frequencies or discontinuities are introduced in the signal - the data filling algorithm is a MATLAB code that is available upon request to F. Lambert. We ensured that only relatively small missing data gaps were filled by using this method, so no large data sections were artificially created. The original and interpolated data are shown in Supplementary figure S1. Since our custom algorithm does not introduce discontinuities in the time series, we used this method for our analysis. In contrast, the receptor model results using the median-based missing data replacement can lose the seasonal signal of some species. Accordingly, the model results using the median-based filling algorithm yielded more variability in Cl, Ti Cr, Ni and As (species with important number of blanks, see Supplementary figure S1).

4) Page 3: A detailed QA/QC section for XRF analysis should be included. How often were the "QC" samples run? (What % age?). No estimates of recovery. What is the limit of quantitation? What is the uncertainty? Any blank correction? Precision and

accuracy?

Answer: A new section on Laboratory and QA/QC analysis was added with the most of this information.

5) Page 4: Did the authors find selenium?

Answer: Selenium was initially considered, but finally removed, because we couldn't get a source apportionment model using Se. Near 27% of Se data were either below LOD or missing; this might explain why we couldn't get a statistically significantly model that included Se.

6) Page 4: Did the authors do the PMF analysis for the missing data? How was this handled?

Answer: The missing data were treated in two separate ways. The first one consisted in leaving them blank and letting the models use their internal algorithm to deal with them, which consists of replacing them with the median values of the complete time-series, for each element. Since replacing missing data with the median can lead to distortions in the data, we also used a custom-written algorithm. This method interpolates up to three consecutive missing values using a piecewise cubic interpolation algorithm. Sections of four or more consecutive missing values are filled by summing up a mirrored copy of equal length of the data on both sides of the empty section, weighted by a cos2 function. We ensured that only relatively small gaps were considered to fill in the missing data, to avoid creating artificial variability in the data. Although both methods yielded comparable results, we have used the custom-written algorithm in this analysis, as it does not introduce discontinuities in the time series.

This methodology is now better explained and we have also added a new Supplementary Figure 1 in the revised manuscript that shows the effect of the filling algorithm.

The missing data in those twenty-two species were dealt with as follows. First, we let the receptor models' (PMF5, UNMIX6) internal algorithms deal with them, which consists of replacing missing values with the median of the complete time-series for each species. Since replacing missing data with the median can lead to severe distortions in the data, we have also used a custom-written algorithm. This method interpolates up to three consecutive missing values using MATLAB's piecewise cubic interpolation algorithm. Sections of four or more consecutive missing values are filled in by summing up a mirrored copy of equal data length on both sides of the missing records, weighted by a cos2 function, thus ensuring that no artificial frequencies or discontinuities are introduced in the signal - the data filling algorithm is a MATLAB code that is available upon request to F. Lambert. We ensured that only relatively small missing data gaps were filled by using this method, so no large data sections were artificially created. The original and interpolated data are shown in Supplementary figure S1. Since our custom algorithm does not introduce discontinuities in the time series, we used this method for our analysis. In contrast, the receptor model results using the median-based missing data replacement can lose the seasonal signal of some species. Accordingly, the model results using the median-based filling algorithm yielded more variability in Cl, Ti Cr, Ni and As (species with important number of blanks, see Supplementary figure S1).

See supplementary Figure S1

7) Page 5: The contribution of Pb from industrial emissions cannot be ruled out. Motor vehicle is not the only source of Pb.

Answer: We agree with the reviewer. More discussion about the Pb from industrial emissions is added.

8) Page 7: "artefact" should be "artifact"

Answer: This was corrected in the revised manuscript

9) Page 8: "Gramsch et al. (2013)". Missing in the reference section.

Answer: This was corrected in the revised manuscript

10) Page 8 Lines 10 – 12: Did the private cars use diesel as a fuel? Primary source of

BC are emissions from diesel engines, cook stoves, wood burning and forest fires.

Answer: The sentence from lines 9-12 is a summary of the conclusions in (Gramsch et al, 2013). After reviewing that paper again, we have decided to drop this reference from that paragraph. The reason is that those authors compared ambient concentrations of BC in June 2005 and June 2007 in several streets (roadside sites). However, monthly precipitations were 173 and 80 mm, respectively, so the ambient BC changes reported by those authors are explained by changes in traffic emissions and meteorological conditions as well.

11) Page 12: "Boisier, J.P., . . .. . ..Mu?????oz, F.," should be corrected. Answer: This has been corrected in the revised manuscript

––––––––––––––––––––––––––

[Figure]

Figure S1 Replacement of missing data. Original data (red line) with missing data filled using a custom-written algorithm (blue line) for Cl, Ti, Cr, Ni, Cu and As are show on the left. The right hand side of the figure shows a zoom of the left panels for selected periods.

[Figure]

**Fig. 1.**

---

## Author Response (AR1)

**General comments**

We thank the reviewers for all the comments and questions that made it possible to vastly improve the manuscripts and the robustness of the results. The main changes to the new manuscript are focused in three areas: i) an appropriate description of the sampling and the quality assurance/quality control of the chemical data, ii) new comparative discussion of our results with other Latin American studies, but mostly with other cities abroad with similar multi-year data and methodology to ours, iii) more discussion about meteorological and climatic phenomena that play an important role on Santiago's air quality.

We also include a revised manuscript with track changes below the point-by-point discussion.

Response to comments by Reviewer 1

1) Overall Assessment This study involved a very large number of samples and for a long time series: 1243 24-hour filter samples of ambient PM2.5 collected between April-1998 to August-2012. It was used two different source-receptor models (PMF 5.0 and Unmix 6.0). The detailed study shows that the main aerosol sources for PM2.5 were: motor vehicles (37%), industrial sources (19%), copper smelters (14%), wood burning (12%), coastal sources (10%), and urban dust (3%).

For a very dry region, it is surprising that urban dust is only 3% of aerosol mass, even considering that the analysis is for PM2.5. Some of the dust factor must have gone in the vehicular source of other factors.

Answer and comment:

The reviewer has pointed out that our estimated urban dust is low for an arid region. However, Santiago is rather a semi-arid zone, for 1998-2012, the annual precipitation had a mean of 320 mm, a low of 110 mm (in 1998) and a high of 620 mm (in 2002). If we compare our concentrations of PMF-resolved soil factor in Santiago (average: 1.1 µg/m$^3$) with those estimated in several cities in California, USA (a region with a very similar climate) we find that:
a) Schauer et al (1996) have reported urban dust between 0.5 and 0.9 µg/m$^3$ (6.8 – 14.3%) in Pasadena, Dowtown LA, West LA and Rubidoux, CA, as an annual average for 1982, using CMB as receptor model.
b) Wang and Hopke (2013) have reported a 10-year source apportionment (PMF) at San Jose, CA and have found an average road dust of 0.58 µg/m$^3$ (5.1%) for 2002-2012
c) Hasheminassab et al (2014) have analyzed ambient PM2.5 at Central Los Angeles and Rubidoux, CA for the period 2002-2013. They have found, using PMF, an average soil contribution of 0.8 – 1.1 µg/m$^3$ at both sites (5 and 6%, respectively).
d) Kim et al (2010) have analyzed data between 2003 and 2005 at the two sites above, using PMF. They have found that the soil contribution varies between 1.5 and 2.0 µg/m$^3$ (6.9 and 9.8%)for Central Los Angeles and 1.6-1.9 µg/m$^3$ (6.0-7.6%) for Rubidoux, CA.

Furthermore, in other long term source apportionment studies carried out using PMF, the PMF resolved soil contribution (in µg/m$^3$) is similar in magnitude: 0.6 for the Sidney Basin between 1998 – 2009 (Cohen et al, 2011), 1.6 for Hanoi, Vietnam between 2001-2008 (Cohen et al, 2010), 0.5 and 0.8 in Detroit and Chicago for 2001-2014 (Milando et al, 2016).

Therefore, our results, on a mass basis, are within the values reported at urban sites with a similar Mediterranean climate as well as in other cities. Nonetheless, we agree with the reviewer that some urban dust may be mixed in with the motor vehicle source.

In the revised manuscript, we have added section 1.1 and two tables (1 and 2):

**1.1 Source apportionment data analyses**

[revised manuscript text omitted]

2) After analyzing the 15 years time series, the results show that over the 15 years, the emissions from motor vehicles, industrial sources, copper smelters, and coastal sources declined by about 21, 39, 81, 59, and 59% respectively, while wood burning didn't change and urban dust increase by 72%. Do you have an estimate for the standard deviation of this important result? The significance of these values depends on the standard deviations that are not reported. Are these reduction numbers all statistically significant at the 95% confidence interval?

Another point is that it is not correct you say that the EMISSIONS were reduced, because you have not measure the emissions, but atmospheric concentrations. I think the best term would be: "The reduction of the impact of the different sources to atmospheric concentrations were". Also important is that there is a lack overall in the whole manuscript of standard deviation for the reported values. Even mean concentrations for PM2.5 do not report their standard deviation. The standard deviation is as important as the average value.

Answer and comment:

We certainly agree with the reviewer on this, so now we quote uncertainties for all reported results. Likewise, the trends that come from robust linear regression of model source contributions against time are now reported with their 95% confidence intervals; in this way, we can conclude about the significance of each one. On the other hand, the standard deviation is not the best parameter to describe dispersion of non-normally distributed data, so we have used the MAD (median absolute deviations) in the revised manuscript. In discussing our results, we now refer to 'impacts' or contributions rather than to 'emissions'.

For example, the following paragraph is from the abstract in the revised manuscript:

PMF resolved six sources that contributed to ambient PM2.5, with UNMIX producing similar results: motor vehicles (37.3±1.1%), industrial sources (18.5±1.3%), copper smelters (14.4±0.8%), wood burning (12.3±1.0%), coastal sources (9.5±0.7%), and urban dust (3.0±1.2%). Our results show that over the 15 years analyzed here, four of the resolved sources significantly decreased [95% Confidence Interval]: motor vehicles 21.3% [2.6, 36.5], industrial sources 39.3% [28.6, 48.4], copper smelters 81.5% [75.5, 85.9], and coastal sources 58.9% [38.5, 72.5], while wood burning didn't significantly change, and urban dust increased by 72% [48.9, 99.9].

3) I feel that in the overall manuscript and also in the reference list, there are very few references to similar studies in other cities. It looks as the study has no connections to other urban areas in Latin America and other places. It looks too isolated in the context on urban aerosol source apportionment. It is important to set the manuscript in a broader context of similar studies done in other urban areas, such as Mexico City, Sao Paulo, La Paz, Quito, etc, as well as some Indian cities that could share similar sources. There is an excess of Chilean studies reported, and a lack of other studies worldwide.

Answer and comment:

We agree with the reviewer. We have now included a revised Table 1 that summarizes source apportionment studies conducted in Latin American cities (WHO, 2017). We have commented on the similarities among them in the introductory section.

We did not find any long-term source apportionment studies in Latin American cities. Therefore, we have summarized long-term source apportionment studies carried out abroad, with an emphasis on California because of the similarities with Santiago's climate. We have added a new Table 2 with comparisons with the following long-term $PM_{2.5}$ source apportionment studies (all carried out using PMF) that we have found in the literature:

a) 2002-2012 in San Jose, CA (Wang and Hopke, 2013)
b) 2002-2013 in Central Los Angeles and Rubidoux, CA (Hasheminassab et al., 2014)
c) 2002-2011 in Detroit and Chicago (Milando et al., 2016)
d) 1998-2009 in Sidney, Australia (Cohen et al., 2011)
e) 2002-2011 in Kuala Lumpur, Malasya (Rahman et al., 2015)
f) 2001-2008 in Hanoi, Vietnam (Cohen et al., 2010)

We have also commented upon these studies in the introduction section (see answer to question 1 above).

4) Figure 2 shows that PMF has not separated residual oil combustion that UNIMIX attributes 7%. There is no discussion on why the two models provided such different results. Of course residual oil combustion must be present in Santiago. In PMF, where Vanadium and Nickel was attributed? This is an important issue that was not discussed in the manuscript.

Answer and comment:

UNMIX resolved the oil combustion as a unique source, apportioning 85% of Ni concentration to that source profile. The PMF solution apportions 56.5 and 19.5% of Ni concentration to the motor vehicles and industrial source profiles, respectively; in other words, PMF mixes sources that come together at the receptor site, transported by winds. This is a consequence of the different methodologies used by PMF and UNMIX to compute source profiles.

We acknowledge that vanadium and nickel are good tracers for oil industrial combustion, but we removed vanadium from the model, because we couldn't obtain a PMF solution using this element. This was due to prolonged periods with vanadium values below LOD between 2002 and 2006. During that period, Santiago's industry used natural gas as industrial fuel, explaining those low vanadium records.

We have added the above two paragraphs in the discussion section.

5) Figures 3 to 9 shows boxplots that are difficult to read, and provide limited information with the outliners. I suggest only shows 50, 75 and 25 percentile, and forget about the outliers, to improve the readability of the figures.

Answer and comment:

We agree with the reviewer. We have improved those figures. As an example, we show below two of those figures.

**Figure 1 Temporal evolution of PM2.5 concentrations in Parque O'Higgins monitoring station in central Santiago. The red line shows the annual median.**

[Figure]

5    **Figure 2 Top panel: Time series of motor vehicles contribution to PM2.5 and the annual median in red. Bottom panel: p-value from a Mann-Whitney hypothesis test comparing the medians of both halves of a sliding window, repeated for 3 different windows lengths (320, 480 and 640 days for blue, red and yellow, respectively).**

[Figure]

6) You discussed the impact of sources to PM$_{2.5}$. What about the meteorology? Did it rain less? more? Cloud cover has changed? Wind direction has changed? Inversions got stronger? Since aerosol concentrations are a function of sources and meteorology, you need to discuss the possible changes in meteorology in detail. I think that the study needs important improvements before it could be considered for publication in ACP. There are several important specific comments that needs to be addressed as well as the
15   general comments discussed above.

Answer and comment:

We do agree, and we have completed the discussion on meteorology. In fact, Central Chile is characterized by significant inter-
20   annual variability in connection with El Niño Southern Oscillation (ENSO) (Garreaud et al., 2009) This characteristic inter-annual variability is illustrated in Supplementary figure 5 below that shows monthly anomalies in precipitation (mm) registered in Santiago downtown since 1960 (Data available at http://explorador.cr2.cl/).

**Figure S5 Monthly precipitation anomalies from the mean in downtown Santiago, 1960-2015. Source: http://explorador.cr2.cl/**

[Figure]

5  We can see from the above figure that there was a downward trend in annual precipitation between 1998 and 2012. Furthermore, since 2007, central and southern Chile has been affected by an extended and persistent drought, partly caused by natural variability and partly linked to a global warming trend ((CR2), 2015; Boisier et al., 2016). We think this drought is a contributing factor in explaining the increase in soil dust contribution in our PMF solution. Likewise, the above downward trend in precipitation implies a worsening of ventilation in Santiago's basin along the period analyzed. However, our trend analysis shows four major $PM_{2.5}$

10  sources decreasing their contributions in the same period. The fact that all our trend estimates for those four sources were negative and significant means that they are conservative estimates, because we did not adjust them for meteorological conditions. The latter computation is beyond the scope of this study because it would require inverse modeling to estimate source strengths.

Regarding mixing height observations, the Chilean Meteorological Service does not launch radiosondes in Santiago, except for

15  limited, short-term campaigns. The only data available are collected with a ceilometer since 2008 (Muñoz et al., 2010; Muñoz and Alcafuz, 2012); these data have significant diurnal, day-to-day, seasonal and possibly inter-annual variability. The following figure illustrates boundary layer height (BLH) retrieved at the Geophysics Department in downtown Santiago between January 1st, 2007 and December 31st 2013 (data kindly provided by Prof. Muñoz). The methodology for the retrievals is described in Muñoz and Undurraga (2010), and considers cloud-free data between 10 and 15 local time (UTC-4). The figure shows a clear seasonality in

20  BLH, with peak values in the austral summer season and lowest values during the austral winter. We see no apparent temporal trend on BLH, so we think this meteorological variable played no role in the temporal trends estimated for all sources resolved by the receptor model analysis.

[Figure]

We added these new comments to the introduction and discussion sections.

7) Specific comments

Page 1 – line 18: the word WERE was missing

Answer: This has been corrected in the revised manuscript

Page 2 line 4 – instead of "impeding horizontal air movements", maybe it is better making it difficult the air mass transport over the metropolitan region.

Answer: This has been corrected in the revised manuscript

Page 2 Line 7 – It would be great to have more information on mixing layer heights than only the expression: The mixing layer shows a marked diurnal cycle (Saide et al., 2011)."". How much is the mixing layer height over winter and summer at midday? Frequency of thermal inversions? Etc. . .

Answer:  This has been corrected in the revised manuscript. Please see the above response regarding meteorological variables in Santiago.

Page 3 line 5 –upperscript for ug/m3.

Answer: This has been corrected in the revised manuscript

Page 4 – Line 33: The detection limit is an important variable, because of the phrase: "We decided to keep only those elements for which more than 70% of the samples contained valid measurements above the detection limit.". There are many ways to derive detection limits in XRF analysis. How were these derived? Blank variability was included? It was derived using 3 sigma? It was derived using multiple analysis of the same filter? How much was the average detection limit in ng/m3 for each element?.

Answer: A new section on Laboratory and QA/QC analysis was added with this information (section 2.2). For each species LOD was calculated as 3 times the standard deviation of field blanks.

Page 4 – Which filter were used? I guess were 37 mm Teflon filters, but this needs to be explicitly mentioned. The same filter was used over the 12 years of sampling?

Answer: A new section on Laboratory and QA/QC analysis was added with this information (section 2.2). 37 mm Teflon filter were used and the methodology was the same throughout the 15 years.

Page 4 The XRF methodology is described under the sampling station section, which is not correct. Suggestion: Open a new section to describe the XRF methodology.

Answer: A new section on Laboratory and QA/QC analysis was added with this information (section 2.2).

Below we present the section on QA/QC included in the revised manuscript that answers the above three specific comments.

**2.2 Laboratory and QA/QC analysis**

Filters were inspected before being used, and the particles' concentration were determined gravimetrically using a microbalance, with a resolution of 0.01 mg. All filter (blank and filter samples) were stored at constant temperature (22±3°C) and relative humidity (40% HR ±3%) for a least 24-hours before being weighed. Those filters were analyzed using X-ray fluorescence (XRF) at the Desert Research Institute, Reno, NV, USA. The Ministry for the Environment provided the database containing the elemental analyses of those filters. In order to build statistical models based on robust chemical signals, we decided to keep only those elements selected in other studies that used the same data (CMM-MMA, 2011; Koutrakis et al., 2005; Sax et al., 2007; Valdes et al., 2012), for which more than 75% of the samples contained valid measurements above the detection limit. The limit of detection

(LOD) was calculated for each element as three times the standard deviation of the blank (blanks represented approximately 10% of the samples). This public database (gravimetry and elemental analysis) has been used in several studies and all of them have already described the laboratory and QA/QC methodology (CMM-MMA, 2011; Jhun et al., 2013; Koutrakis et al., 2005; Sax et al., 2007). Thus, out of the 49 elements reported, 22 agreed with the criteria described above (Na, Mg, Al, Si, S, Cl, K, Ca, Ti, V,

5 Cr, Mn, Fe, Ni, Cu, Zn, As, Se, Br, Sr, Ba and Pb). Out of these, some were discarded because of large data gaps in the time series (Mg, V, Sr, Ba, Se), suspicious sources (Pb and Br, see discussion below), or because the model was not significant (Na and Ca). In the end, 12 elements were used for our analysis: Al, Si, S, Cl, K (as Kns), Ti, Cr, Fe, Ni, Cu, Zn, As. We used two separate methods to address the small missing data gaps in the time series of the selected elements. First, we let the receptor models' (PMF5, UNMIX6) internal algorithms deal with them, which consists of replacing missing values with the median of the complete time-

10 series for each species. Since replacing missing data with the median can lead to severe distortions in the data, we have also used a custom-written algorithm. This method interpolates up to three consecutive missing values using MATLAB's piecewise cubic interpolation algorithm. Sections of four or more consecutive missing values are filled in by summing up a mirrored copy of equal data length on both sides of the missing records, weighted by a $\cos^2$ function, thus ensuring that no artificial frequencies or discontinuities are introduced in the signal — the data filling algorithm is a MATLAB code that is available upon request to F.

15 Lambert. None of the species selected for the receptor model analysis features any data gap larger than 10 data points. Species with larger gaps were ultimately discarded from the analysis. The original and interpolated data are shown in Supplementary figure S1. Since our custom algorithm does not introduce discontinuities in the time series, we used this method for our analysis. In contrast, the receptor model results using the median-based missing data replacement can lose the seasonal signal of some species. Accordingly, the model results using the median-based filling algorithm yielded more variability in Cl, Ti Cr, Ni and As (species

20 with important number of blanks, see Supplementary figure S1).

Page 5 – Treatment of missing values: Again: This is described under the section sampling, and this is not appropriate. The treatment of missing values (up to 30% of the samples) is important and needs better description. Substitute the missing values by the median is certainly not appropriated, and I am surprised to see that the results using this wrong procedure and interpolation

25 using better algorithms provide similar results. I really do not believe that this is the case. As up to 30% of data for some variables was artificially introduced in the analysis, a much better discussion on the effects must be provided in the manuscript.

Answer: We actually discarded all species that featured large data gaps from our analysis. We only used 12 species that were relatively compete and feature only small data gaps. This is why the results were not very different between the two methods. We

30 have included a more thorough description in the revised manuscript, as follows:

The missing data in those twenty-two species were dealt with as follows. First, we let the receptor models' (PMF5, UNMIX6) internal algorithms deal with them, which consists of replacing missing values with the median of the complete time-series for each species. Since replacing missing data with the median can lead to severe distortions in the data, we have also used a custom-

35 written algorithm. This method interpolates up to three consecutive missing values using MATLAB's piecewise cubic interpolation algorithm. Sections of four or more consecutive missing values are filled in by summing up a mirrored copy of equal data length on both sides of the missing records, weighted by a $\cos^2$ function, thus ensuring that no artificial frequencies or discontinuities are introduced in the signal — the data filling algorithm is a MATLAB code that is available upon request to F. Lambert. We ensured that only relatively small missing data gaps were filled by using this method, so no large data sections were artificially created.

40 The original and interpolated data are shown in Supplementary figure S1. Since our custom algorithm does not introduce discontinuities in the time series, we used this method for our analysis. In contrast, the receptor model results using the median-based missing data replacement can lose the seasonal signal of some species. Accordingly, the model results using the median-based filling algorithm yielded more variability in Cl, Ti Cr, Ni and As (species with important number of blanks, see Supplementary figure S1).

In the supplementary file we have included the following figure:

**Figure S1 Example of replacement of missing data. Original data for Cl (in blue) with missing data filled using a custom-written algorithm (in red). Shown are a) Cl concentration on a logarithmic scale, b) zoom of a data range with small and large filled data gaps**

[Figure]

Page 5 Line 17: You need to define very precisely what you mean by BEST Model in the phrase: "We considered 13 species that yield the best model: Al, Si, S, Cl, non-soil K (Kns), Ti, Cr, Mn, Fe, Ni, Cu, Zn and As".

Answer: The "best model" means the model that has the best regression parameters and at the same time the highest number of resolved sources (statistically significant regression coefficients). We have added the next sentence:

"*This model had the most robust regression parameter and the highest number of statistically significant source factors*"

Page 5 lines 25-27. The factor 0.3 relating K to Fe as in the methodology of Lewis et al., can change a lot from site to site. You mentioned that you have done a regression, but you certainly needs much better explanation.

Answer: Indeed, the factor 0.3 is specific for the sampling site and was calculated with the data collected therein. We have added a supplementary figure with the K-Fe scatter plot showing a lower edge with a 0.3 slope (see figure in our next answer below).

You added a new variable that is not statistically independent from the others. This can bring problems in multivariate models. This needs to be much better discussed and explained.

Answer: We knew about this problem, so we have added Kns in and removed K from the model. We have added the following sentence to the manuscript:

*To provide a tracer associated with wood burning, we have added the non-soil potassium parameter Kns calculated as Kns=(K-0.3•Fe) and removed K from the model. The 0.3 coefficient was obtained from a K-Fe edge plot (supplementary figure 2)"*

**Figure S2 K-Fe edge plot. To provide a tracer associated with wood burning we added the non-soil potassium parameter Kns calculated as Kns=K-0.3•Fe, from the K-Fe edge plot**

[Figure]

5   Page 6 line 3: Even with a very high number of samples, you have NOT explained the origin of 25% of the variance of the PM2.5. Why you only explained 75% of the variance? This is a low value for multivariate analysis from urban areas. This is mentioned as " PMF 5.0 produces a six factors solution that explained 74% of the variance in ambient PM2.5 " You have not discussed this important point. It is strange that the unexplained PM2.5 is only 5-7% in Figure 2, and you explained only 75% of the variability.

10   Answer: The average source contributions must add up to the average PM$_{2.5}$ concentration by design (except by an intercept value). On the other hand, model variability stands for the scatter of individual (daily) model estimates compared with the actual observed (daily) PM$_{2.5}$. These are different parameters. It is difficult to find receptor models that explain more than 90% of the variance. In our case, the unexplained 25% may be ascribed to the following causes:

15     i)      Actual source profiles do not stay constant over the whole modeling period, adding uncertainty to the model results.
    ii)     We could not include organic/elemental carbon into the model. The lack of these two components may increase uncertainty in the resolved source profiles.  For instance, the ratio OC/EC is helpful in discriminating motor vehicles from wood burning.
    iii)    We only had inorganic tracer species in the dataset, which put a limitation on our analysis. We knew that secondary
20             organic aerosols (SOA) may be relevant in the warm season (October – April), as shown by Villalobos et al (2015) for Santiago in 2013 (application of CMB with organic molecular markers); these authors estimated that nearly 30% of total PM$_{2.5}$ was identified as SOA.

We have added these new comments in the discussion section.

Page 7 – Line 13: There is a very important lack overall in the whole manuscript of standard deviation for the reported values. Even mean concentrations for PM2.5 do not report their standard deviation. This is unacceptable in science: All reported average values needs to have their standard deviation reported together with the mean value. For instance in the phrase: "Over the whole study period, the daily mean (24 h)
30   concentration of PM2.5 was 35.60 µg/m3 and the median 24.19 µg/m3"

Answer: We agree with this comment. Since data have a non-normal distribution, we have used the median absolute deviation (MAD) as a measure of dispersion for each reported median and mean value.

35   Page 8 line 20: Very strange that the reduction in industrial sources was attributed to decrease in sulfur in the DIESEL, used in the transportation. This needs to be correctly explained. This is on the phrase: "In Figure 5 we show temporal evolution of the source identify by PMF as industrial sources. This source reduced its 20 contributions from 1998 to 2012 by 2.63 µg/m3 (39.23%, p=0.11x10-8). This improvement can be explained by the reduction policies for sulfur in diesel fuel".

40   Answer: The quality of industrial diesel fuel was also improved. We put in the page 8 (3.3.2 section) the following new sentence: "[…], that can be explained by a reduction of sulfur in industrial diesel, which was reduced from 1000 to 300 ppm in 2001 […]"
5    General: The project seems to be carefully thought out. The analytical methodology (PMF 5.0 and Unmix 6.0) seems appropriate; however, a separate detailed sampling and QA/QC section is needed. Language and spellings need to be improved. Concentrations should be expressed in 3 significant figures throughout the text and in the figures and tables. The author should compare the data with other studies in urban areas. As such I recommend that it be published with major revision:

1) Page 3: "µg/m3" should be "µg/m3" - be consistent throughout the text, figures, and tables.

10    Answer: This was corrected in the revised manuscript

2) Page 3: "24-hour" or "24-hours" or 24 h" – be consistent with one of them.

Answer: This was corrected in the revised manuscript

3) Page 3: No mention for the sampling and analysis for PM2.5? How PM2.5 samples were obtained? Which type of filter was used? Were the filters weighed in the clean room? Which analytical balance was used? Any QA/QC?

15    Answer: the follow section was added with this information.

**2.2 Laboratory and QA/QC analysis**

Filters were inspected before being used, and the particles' concentration were determined gravimetrically using a microbalance, with a resolution of 0.01 mg. All filter (blank and filter samples) were stored at constant temperature (22±3°C) and relative humidity (40% HR ±3%) for a least 24-hours before being weighed. Those filters were analyzed using X-ray fluorescence (XRF)

20    at the Desert Research Institute, Reno, NV, USA. The Ministry for the Environment provided the database containing the elemental analyses of those filters. In order to build statistical models based on robust chemical signals, we decided to keep only those elements selected in other studies that used the same data (CMM-MMA, 2011; Koutrakis et al., 2005; Sax et al., 2007; Valdes et al., 2012), for which more than 75% of the samples contained valid measurements above the detection limit. The limit of detection (LOD) was calculated for each element as three times the standard deviation of the blank (blanks represented approximately 10%

25    of the samples). This public database (gravimetry and elemental analysis) has been used in several studies and all of them have already described the laboratory and QA/QC methodology (CMM-MMA, 2011; Jhun et al., 2013; Koutrakis et al., 2005; Sax et al., 2007). Thus, out of the 49 elements reported, 22 agreed with the criteria described above (Na, Mg, Al, Si, S, Cl, K, Ca, Ti, V, Cr, Mn, Fe, Ni, Cu, Zn, As, Se, Br, Sr, Ba and Pb). Out of these, some were discarded because of large data gaps in the time series (Mg, V, Sr, Ba, Se), suspicious sources (Pb and Br, see discussion below), or because the model was not significant (Na and Ca).

30    In the end, 12 elements were used for our analysis: Al, Si, S, Cl, K (as Kns), Ti, Cr, Fe, Ni, Cu, Zn, As. We used two separate methods to address the small missing data gaps in the time series of the selected elements. First, we let the receptor models' (PMF5, UNMIX6) internal algorithms deal with them, which consists of replacing missing values with the median of the complete time-series for each species. Since replacing missing data with the median can lead to severe distortions in the data, we have also used a custom-written algorithm. This method interpolates up to three consecutive missing values using MATLAB's piecewise cubic

35    interpolation algorithm. Sections of four or more consecutive missing values are filled in by summing up a mirrored copy of equal data length on both sides of the missing records, weighted by a $\cos^2$ function, thus ensuring that no artificial frequencies or discontinuities are introduced in the signal — the data filling algorithm is a MATLAB code that is available upon request to F. Lambert. None of the species selected for the receptor model analysis features any data gap larger than 10 data points. Species with larger gaps were ultimately discarded from the analysis. The original and interpolated data are shown in Supplementary figure

S1. Since our custom algorithm does not introduce discontinuities in the time series, we used this method for our analysis. In contrast, the receptor model results using the median-based missing data replacement can lose the seasonal signal of some species. Accordingly, the model results using the median-based filling algorithm yielded more variability in Cl, Ti Cr, Ni and As (species with important number of blanks, see Supplementary figure S1).

4) Page 3: A detailed QA/QC section for XRF analysis should be included. How often were the "QC" samples run? (What % age?). No estimates of recovery. What is the limit of quantitation? What is the uncertainty? Any blank correction? Precision and accuracy?

Answer: A new section on Laboratory and QA/QC analysis was added with the most of this information.

5) Page 4: Did the authors find selenium?

10 Answer: Selenium was initially considered, but finally removed, because we couldn't get a source apportionment model using Se. Near 27% of Se data were either below LOD or missing; this might explain why we couldn't get a statistically significantly model that included Se.

6) Page 4: Did the authors do the PMF analysis for the missing data? How was this handled?

Answer: We only selected 12 species for our analysis that only feature small data gaps. Any species with large data gaps was

15 discarded from the analysis. The missing data were treated in two separate ways. The first one consisted in leaving them blank and letting the models use their internal algorithm to deal with them, which consists of replacing them with the median values of the complete time-series, for each element. Since replacing missing data with the median can lead to distortions in the data, we also used a custom-written algorithm. This method interpolates up to three consecutive missing values using a piecewise cubic interpolation algorithm. Sections of four or more consecutive missing values are filled by summing up a mirrored copy of equal

20 length of the data on both sides of the empty section, weighted by a $\cos^2$ function. We ensured that only relatively small gaps were considered to fill in the missing data, to avoid creating artificial variability in the data. The original and interpolated data are shown in Supplementary figure S1. Since our custom algorithm does not introduce discontinuities in the time series, we used this method for our analysis. In contrast, the receptor model results using the median-based missing data replacement can lose the seasonal signal of some species. Accordingly, the model results using the median-based filling algorithm yielded more variability in Cl, Ti

25 Cr, Ni and As (species with important number of blanks, see Supplementary figure S1).

In the supplementary file we have included the following figure

**Figure S3 Example of replacement of missing data. Original data for Cl (in blue) with missing data filled using a custom-written algorithm (in red). Shown are a) Cl concentration on a logarithmic scale, b) zoom of a data range with small and large filled data gaps**

[Figure]

7) Page 5: The contribution of Pb from industrial emissions cannot be ruled out. Motor vehicle is not the only source of Pb.

Answer: We agree with the reviewer. More discussion about the Pb from industrial emissions is added.

8) Page 7: "artefact" should be "artifact"

Answer: This was corrected in the revised manuscript

9) Page 8: "Gramsch et al. (2013)". Missing in the reference section.

Answer: This was corrected in the revised manuscript

10) Page 8 Lines 10 – 12: Did the private cars use diesel as a fuel? Primary source of BC are emissions from diesel engines, cook stoves, wood burning and forest fires.

Answer: The sentence from lines 9-12 is a summary of the conclusions in (Gramsch et al, 2013). After reviewing that paper again, we have decided to drop this reference from that paragraph. The reason is that those authors compared ambient concentrations of BC in June 2005 and June 2007 in several streets (roadside sites). However, monthly precipitations were 173 and 80 mm, respectively, so the ambient BC changes reported by those authors are explained by changes in traffic emissions *and* meteorological conditions as well.

11) Page 12: "Boisier, J.P., . . .. . ..Mu?????oz, F.," should be corrected.

Answer: This has been corrected in the revised manuscript

__Revised manuscript with Trach Changes__

**Temporal evolution of main ambient PM$_{2.5}$ sources in Santiago, Chile, from 1998 to 2012**

Francisco Barraza[1,4], Fabrice Lambert[1,4], Héctor Jorquera[2,5], Ana M.María Villalobos[2], Laura Gallardo[3,4]

[1] Geography Institute, Pontificia Universidad Católica de Chile, Santiago, 7820436, Chile
5  [2] Department of chemical engineeringChemical Engineering and bioprocessesBioprocesses, Pontificia Universidad Católica de Chile, Santiago, 7820436, Chile
[3] Department of Geophysics, Universidad de Chile, Santiago, Chile
[4] Center for Climate and Resilience Research, University of Chile, Santiago, Chile
[5] Center for Sustainable Urban Development (CEDUSCEDEUS), Pontificia Universidad Católica de Chile, Santiago, 7820436,
10  Chile

*Correspondence to*: Francisco Barraza (fjbarraz@uc.cl)

**Abstract.**

The inhabitants of Santiago in, Chile have been exposed to harmful levels of air pollutants for decades. The city's poor air quality is a result of sustained emissionssteady economic growth, and stable atmospheric conditions, averse adverse to mixing and
15  ventilation and favorable forthat favor the formation of oxidants and secondary aerosols. Identifying and quantifying the sources that contribute to the ambient levels of pollutants is key for designing adequate mitigation measures. Knowledge aboutEstimating the temporal evolution of the contribution of each source contributions to ambient pollution levels is also paramount to evaluateevaluating the effectiveness of pollution reduction measures that have been implemented inover the past decades. Here, we quantify the main sources that have contributed to fine particulate matter (PM$_{2.5}$) between April 1998 and August 2012 in
20  Santiago's center downtown Santiago by using two different source-receptor models (PMF 5.0 and UnmixUNMIX 6.0), that rewere applied to elemental measurements onof 1243 24-hour filter samples of ambient PM$_{2.5}$ collected between April 1998 to August 2012. Both models resolve. PMF resolved six sources that contributecontributed to ambient PM$_{2.5}$, with UNMIX producing similar results: motor vehicles (37.3±1.1%), industrial sources (1918.5±1.3%), copper smelters (14.4±0.8%), wood burning (12.3±1.0%), coastal sources (109.5±0.7%), and urban dust (3.0±1.2%). Our results show that over the 15 years analyzed here,
25  four of the emissions fromresolved sources significantly decreased [95% Confidence Interval]: motor vehicles, 21.3% [2.6, 36.5], industrial sources, 39.3% [28.6, 48.4], copper smelters, 81.5% [75.5, 85.9], and coastal sources declined by about 21, 39, 81, 59, and 59% respectively,58.9% [38.5, 72.5], while wood burning didn't significantly change, and urban dust increaseincreased by 72%.% [48.9, 99.9]. These changes are consistent with emission reduction measures, such as improved vehicle andemission standards, cleaner smelting technology, introduction of low sulfur fueldiesel for vehicles and natural gas for industrial processes,
30  emission controls for vehicles, public transport improvements etc.. However, it is also apparent that the mitigation expected from improved public transport, vehicle technology, and fuelthe above regulations has been largely nullifiedpartially offset by the ever-rising numberincreasing amount of private vehicle journeysuse in the past decade. As a consequencecity, 
[revised manuscript text omitted]